# Genetic interactions of G-quadruplexes in humans

Katherine G Zyner[1†], Darcie S Mulhearn[1†], Santosh Adhikari[2],
Sergio Martínez Cuesta[1], Marco Di Antonio[2], Nicolas Erard[1], Gregory J Hannon[1],
David Tannahill[1], Shankar Balasubramanian[1,2,3]*

[1]Cancer Research United Kingdom Cambridge Institute, Cambridge, United Kingdom; [2]Department of Chemistry, University of Cambridge, Cambridge, United Kingdom; [3]School of Clinical Medicine, University of Cambridge, Cambridge, United Kingdom

**Abstract** G-quadruplexes (G4) are alternative nucleic acid structures involved in transcription, translation and replication. Aberrant G4 formation and stabilisation is linked to genome instability and cancer. G4 ligand treatment disrupts key biological processes leading to cell death. To discover genes and pathways involved with G4s and gain mechanistic insights into G4 biology, we present the first unbiased genome-wide study to systematically identify human genes that promote cell death when silenced by shRNA in the presence of G4-stabilising small molecules. Many novel genetic vulnerabilities were revealed opening up new therapeutic possibilities in cancer, which we exemplified by an orthogonal pharmacological inhibition approach that phenocopies gene silencing. We find that targeting the WEE1 cell cycle kinase or USP1 deubiquitinase in combination with G4 ligand treatment enhances cell killing. We also identify new genes and pathways regulating or interacting with G4s and demonstrate that the DDX42 DEAD-box helicase is a newly discovered G4-binding protein.
DOI: https://doi.org/10.7554/eLife.46793.001

*For correspondence:
sb10031@cam.ac.uk

[†]These authors contributed equally to this work

## Introduction

G-quadruplex secondary structures (G4s) form in nucleic acids through the self-association of guanines (G) in G-rich sequences to form stacked tetrad structures (reviewed in *Bochman et al., 2012*; *Rhodes and Lipps, 2015*). In the human genome, over 700,000 G4s have been detected in vitro (*Chambers et al., 2015*). Sequences encoding G4s are enriched in regulatory regions consistent with roles in transcription and RNA regulation (*Huppert and Balasubramanian, 2007*; *Huppert, 2008*), and their over-representation in oncogene promoters, such as *MYC*, *KRAS* and *KIT*, suggests that they are important in cancer and are potential therapeutic targets (reviewed in *Balasubramanian et al., 2011*). Computationally predicted G4s have also been linked to replication origins (*Besnard et al., 2012*) and telomere homeostasis (reviewed in *Neidle, 2010*). In the transcriptome, more than 3000 mRNAs have been shown to contain G4 structures in vitro, particularly at 5' and 3' UTRs, suggestive of roles in posttranscriptional regulation (*Bugaut and Balasubramanian, 2012*; *Kwok et al., 2016*).

G4-specific antibodies have been used to visualise G4s in protozoa (*Schaffitzel et al., 2001*) and mammalian cells (*Biffi et al., 2013*; *Henderson et al., 2014*; *Liu et al., 2016*). More G4s are detected in transformed versus primary cells, and in human stomach and liver cancers compared to non-neoplastic tissues, supporting an association between G4 structures and cancer (*Biffi et al., 2014*; *Hänsel-Hertsch et al., 2016*). More recently, ChIP-seq was used to map endogenous G4 structure formation in chromatin revealing a link between G4s, promoters and transcription (*Hänsel-Hertsch et al., 2016*). G4s are found predominately in nucleosome-depleted chromatin within

promoters and 5' UTRs of highly transcribed genes, including cancer-related genes and regions of somatic copy number alteration. G4s may therefore be part of a regulatory mechanism to switch between different transcriptional states. At telomeres, tandem G4-repeat structures also may help protect chromosome ends by providing binding sites for shelterin complex components (reviewed in *Brázda et al., 2014*). As G4 structures can pause or stall polymerases, they must be resolved by helicases to allow replication and transcription to proceed. Several helicases, including WRN, BLM, PIF1, DHX36 and RTEL1, have been shown to unwind G4-structures in vitro (*Brosh, 2013*; *Mendoza et al., 2016*), and it is notable that fibroblasts from Werner (WRN) and Bloom (BLM) syndrome patients, who are predisposed to cancer, show altered gene expression that correlates with sites with potential to form G4s (*Damerla et al., 2012*).

Small molecules that selectively bind and stabilise G4 formation in vitro have been used to probe G4 biological function. G4 ligands, such as pyridostatin (PDS), PhenDC3 and TMPyP4, can reduce transcription of many genes harbouring a promoter G4, including oncogenes such as *MYC,* in multiple cancer cell lines (*Halder et al., 2012*; *McLuckie et al., 2013*; *Neidle, 2017*). G4-stabilising ligands also interfere with telomere homeostasis by inducing telomere uncapping/DNA damage through the inhibition of telomere extension by telomerase leading to senescence or apoptosis (reviewed in *Neidle, 2010*). 5' UTR RNA G4 structures may also be involved in eIF4A-dependent oncogene translation (*Wolfe et al., 2014*) and their stabilisation by G4-ligands can inhibit translation in vitro (*Bugaut and Balasubramanian, 2012*). Identification of several RNA G4-interacting proteins (reviewed in *Cammas and Millevoi, 2016*), including DEAD/DEAH helicases such as DDX3X, and DHX36 (*Chen et al., 2018*; *Herdy et al., 2018*) additionally suggests specific roles for G4 structures in RNA.

Some G4-stabilising ligands cause a DNA damage-response (DDR); for example, DNA damage sites induced by PDS in human lung fibroblasts mapped to genomic regions at G4s within several oncogenes including *SRC* (*Rodriguez et al., 2012*). Subsequent studies demonstrated that homologous recombination (HR) repair deficiencies can be exploited to selectively kill BRCA1/2-deficient cancer cells with G4 ligands (*McLuckie et al., 2013*; *Zimmer et al., 2016*). Recently, this concept has been applied to BRCA1/2-deficient breast cancers using CX-5461, a G4 ligand currently in clinical trials (*Xu et al., 2017*) (NCT02719977 ClinicalTrials.gov). Overall, these initial studies demonstrate that specific genotypes can be selectively vulnerable to G4-stabilisation and raises the question as to what other genotypes might provide further such opportunities.

We set out to address two main questions (*Figure 1*): 1) which human genes and cellular pathways interact with G4s and 2) what genetic backgrounds selectively lead to enhanced cell killing in the presence of G4 stabilising ligands? We employed PDS and PhenDC3 as representative G4 ligands as these are chemically and structurally dissimilar, but each shows a broad specificity for different G4 structural variants. Both ligands have been widely used as G4-targeting probes in biophysical (*De Cian et al., 2007b*; *Rodriguez et al., 2008*) and biological studies in which they have been shown to impart transcriptional inhibition, telomere dysfunction and replication stalling (*De Cian et al., 2007a*; *Halder et al., 2012*; *Mendoza et al., 2016*).

## Results

### Identification of genetic vulnerabilities to G4-ligands via genome-wide screening

An unbiased genome-wide shRNA screen was performed in A375 human melanoma cells to globally evaluate genetic vulnerabilities to G4-ligands and to identify genes and pathways involved with G4-structures (*Figure 2A*). For this, the pyridine-2,6-bis-quinolino-dicarboxamide derivative, PDS (*Rodriguez et al., 2012*), and bisquinolinium compound, PhenDC3 (*De Cian et al., 2007b*) were chosen (*Figure 2B*). We used the latest generation shERWOOD-Ultramir shRNA pLMN retroviral library, comprising 132,000 shRNAs across 12 randomised pools targeting the protein coding genome, with an average of five optimised hairpins per gene (*Figure 2C*) (*Knott et al., 2014*). A375 melanoma cells were used due to their rapid doubling, stable ploidy and success in other shRNA-dropout screens (*Sims et al., 2011*); they are TP53 wild-type and driven by oncogenic *BRAF* (V600E) and *CDKN2A* loss (*Forbes et al., 2015*). *Figure 2D* outlines our shRNA screening strategy. To identify shRNAs that are lost between the initial (t0) and final (fF) timepoints, unique 3'-antisense

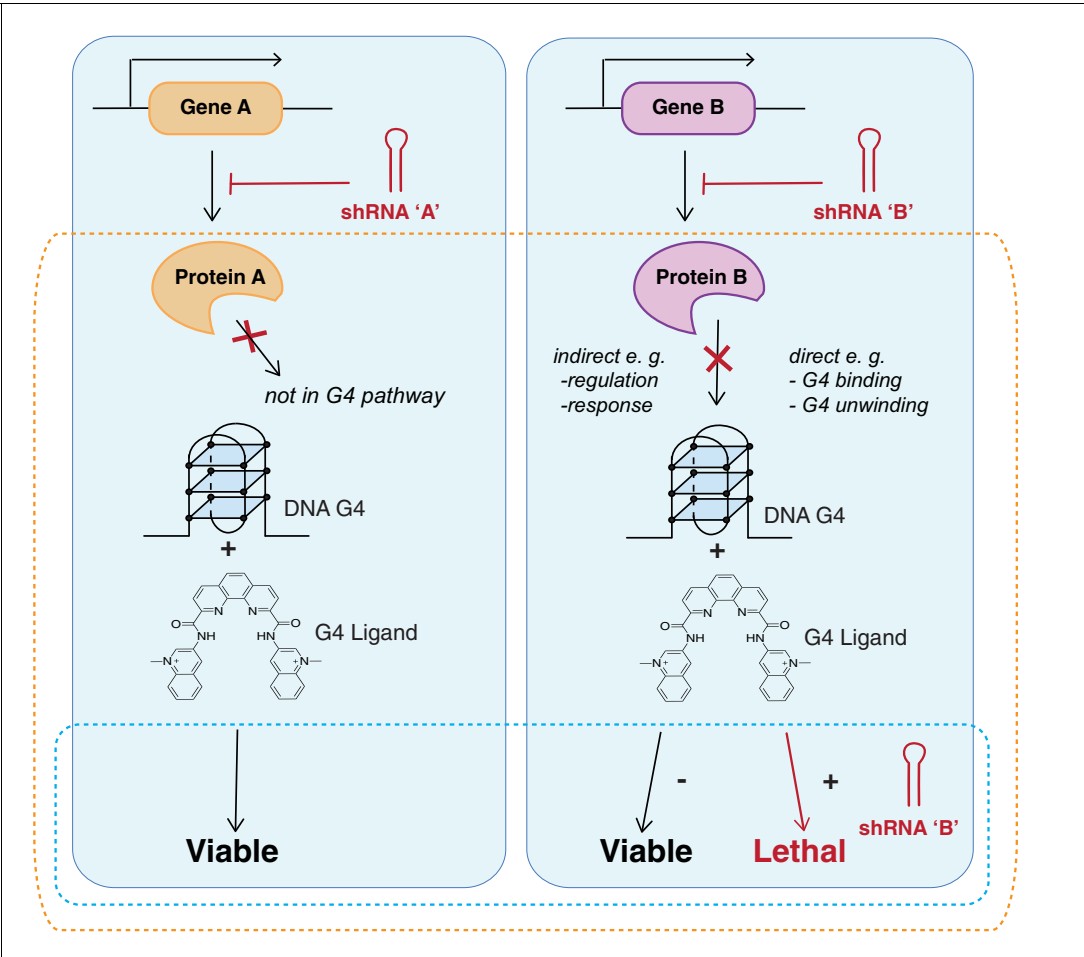

**Figure 1.** Strategy identifying genetic vulnerabilities involved with G4 biology. Genome-wide shRNA silencing combined with G4 structure stabilisation by small molecules identifies genes that when depleted compromise cell viability. Cells are infected with a genome-wide pool of shRNA lentiviruses targeting the protein coding genome followed by G4 ligand treatment to stabilise genomic and/or RNA G4 structures. Two general outcomes are possible: a gene is not required in a G4-dependent process so there is no effect on cell viability (left); or gene silencing results in cell death either due to loss of a direct G4 interaction (e.g. binding/unwinding) or indirectly through gene loss in a G4-dependent pathway (right). In absence of ligand, cells are viable in presence of the shRNA. Dotted boxes highlight genotypes of disease significance for possible G4-based therapies (blue) and genes and biological pathways that involve and/or interact with G4 structures (orange).

DOI: https://doi.org/10.7554/eLife.46793.002

sequences were recovered by PCR and quantified by sequencing. If a gene knockdown compromises cell viability then the associated shRNA will be depleted compared to those targeting non-essential genes: the tF sequence count will be less than t0 thus $\log_2$ fold change (FC, tF/t0) is negative. A pilot using one shRNA pool established that a tF of 15 population doublings can be used to reveal significant G4-ligand-mediated changes [false discovery rate (FDR) $\leq$ 0.05] in shRNA levels using a ligand concentration resulting in 20% cell death ($GI_{20}$, see Materials and methods and *Figure 2—figure supplement 1* for details).

To understand the complete spectrum of G4 vulnerabilities, we first considered the combined set of sensitivities to PDS and PhenDC3 together. For the whole library, when individual shRNAs are considered 9509 (~7%) G4-ligand-specific hairpins (i.e. those not in DMSO) were found to be depleted (FDR $\leq$ 0.05; $\log_2$ FC <0, *Figure 3A*, *Supplementary file 1*). We then reasoned, for a gene knockdown to have compromised cell growth, that a minimum of either 50% or three shRNA hairpins should be significantly depleted for that gene (median $\log_2$ FC <0). This resulted in the identification of 843 G4 ligand-specific gene knockdowns not present in DMSO (*Figure 3B*). We then denoted a more stringent preliminary list of 758 G4 sensitisers as those having a median $\log_2$ FC $\leq$

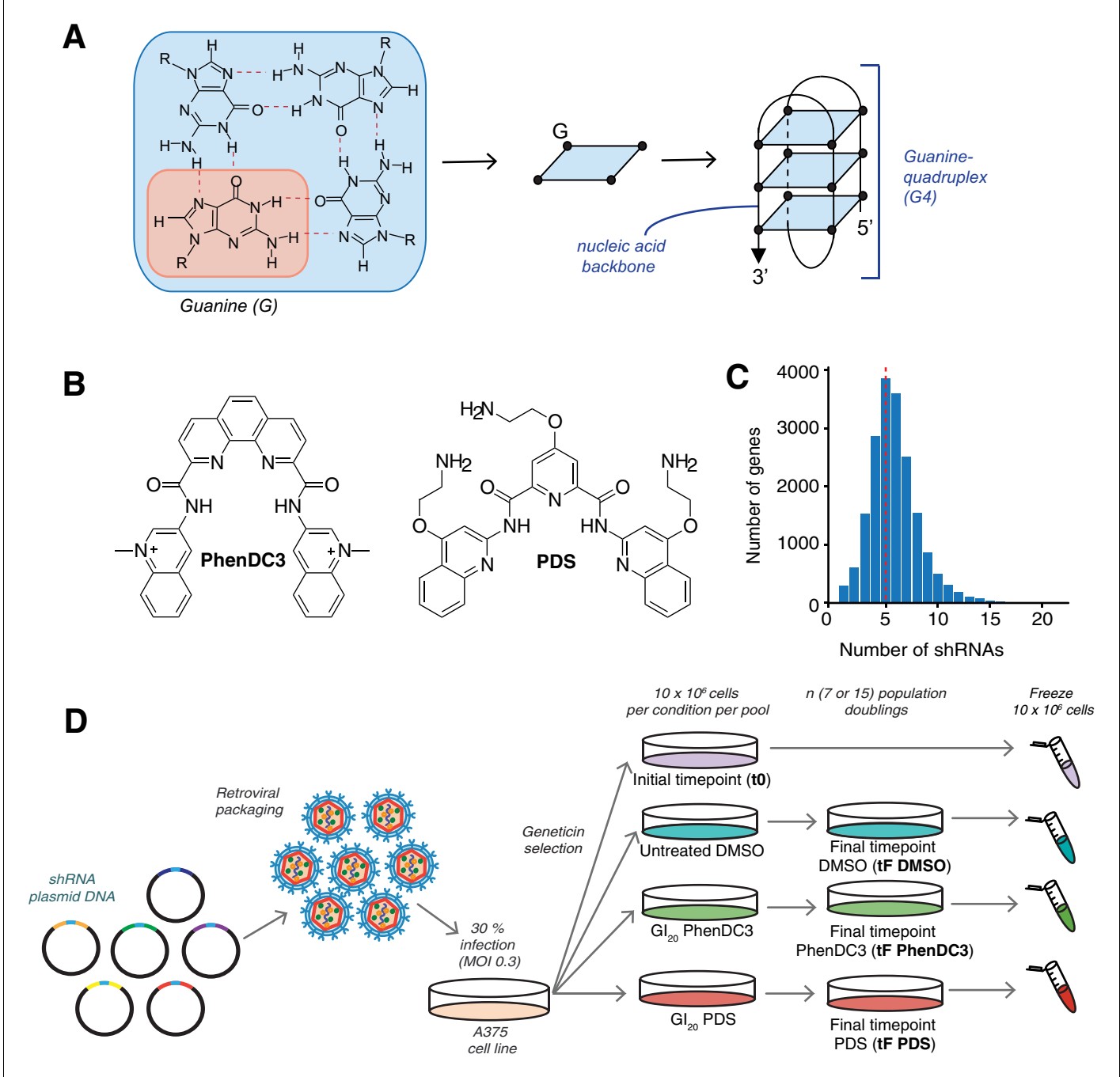

**Figure 2.** shRNA screening pipeline to uncover genetic vulnerabilities to G4 stabilisation. (**A**) A G-tetrad with four interacting guanines (left), which stack to form G4 structures (right). (**B**) Structures of the G4-stabilising small molecule ligands PDS and PhenDC3. (**C**) Distribution of the numbers of shRNAs targeting each gene, with the average indicated by a red dotted line. (**D**) Overall screening approach illustrated for one library pool. Plasmids are retrovirally packaged and A375 cells are infected at multiplicity of infection (MOI) of 0.3 (30%). Following antibiotic selection, an initial time point (t0) is harvested and then cells are cultured for 'n' population doublings in DMSO, PDS or PhenDC3 before the final time point was harvested (tF).
DOI: https://doi.org/10.7554/eLife.46793.003

The following figure supplement is available for figure 2:

**Figure supplement 1.** Genome-wide shRNA screen parameter optimisation.
DOI: https://doi.org/10.7554/eLife.46793.004

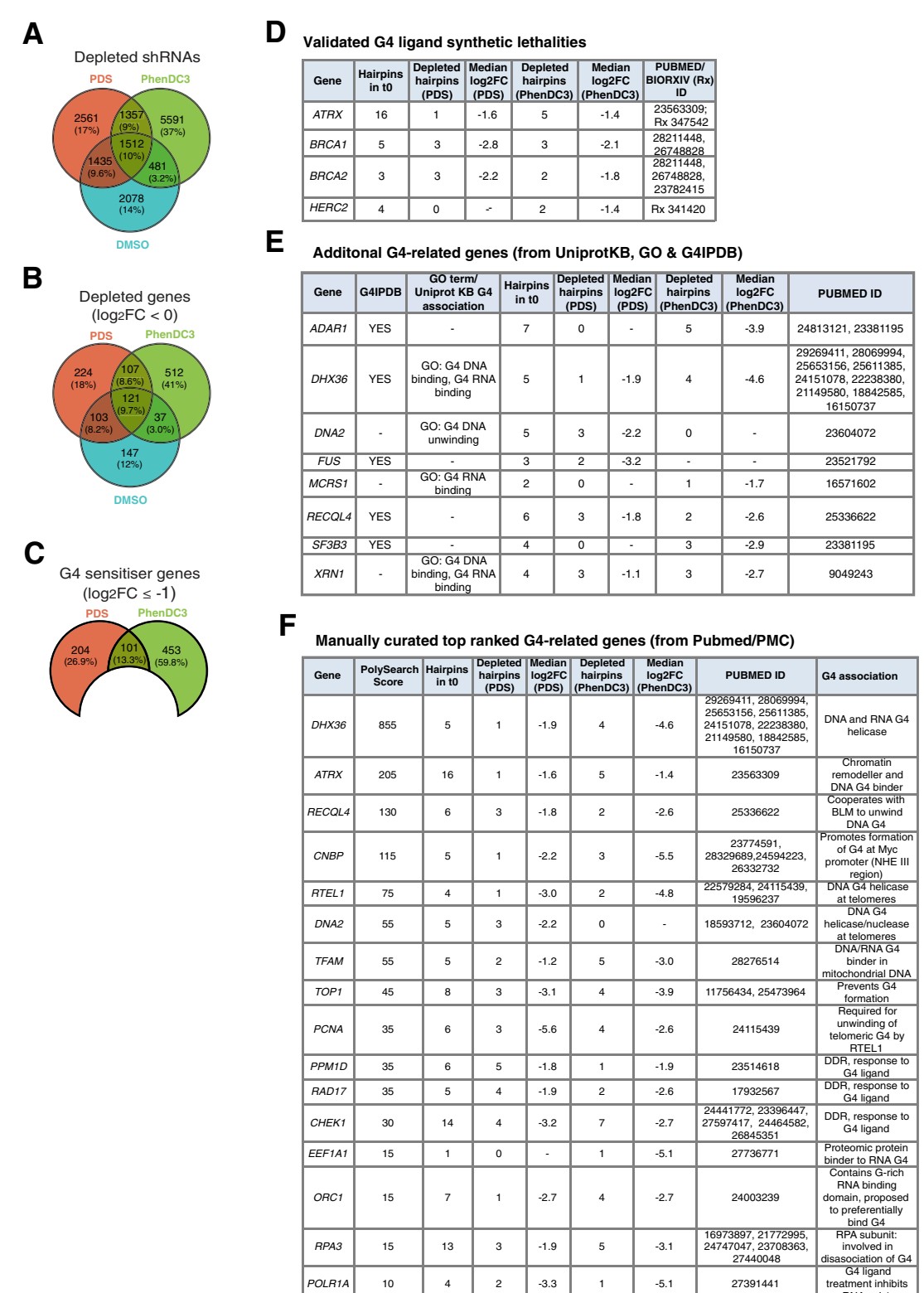

**A** Depleted shRNAs

**B** Depleted genes (log2FC < 0)

**C** G4 sensitiser genes (log2FC ≤ -1)

**D** Validated G4 ligand synthetic lethalities

| Gene | Hairpins in t0 | Depleted hairpins (PDS) | Median log2FC (PDS) | Depleted hairpins (PhenDC3) | Median log2FC (PhenDC3) | PUBMED/ BIORXIV (Rx) ID |
|------|------|------|------|------|------|------|
| ATRX | 16 | 1 | -1.6 | 5 | -1.4 | 23563309; Rx 347542 |
| BRCA1 | 5 | 3 | -2.8 | 3 | -2.1 | 28211448, 26748828 |
| BRCA2 | 3 | 3 | -2.2 | 2 | -1.8 | 28211448, 26748828, 23782415 |
| HERC2 | 4 | 0 | - | 2 | -1.4 | Rx 341420 |

**E** Additonal G4-related genes (from UniprotKB, GO & G4IPDB)

| Gene | G4IPDB | GO term/ Uniprot KB G4 association | Hairpins in t0 | Depleted hairpins (PDS) | Median log2FC (PDS) | Depleted hairpins (PhenDC3) | Median log2FC (PhenDC3) | PUBMED ID |
|------|------|------|------|------|------|------|------|------|
| ADAR1 | YES | - | 7 | 0 | - | 5 | -3.9 | 24813121, 23381195 |
| DHX36 | YES | GO: G4 DNA binding, G4 RNA binding | 5 | 1 | -1.9 | 4 | -4.6 | 29269411, 28069994, 25653156, 25611385, 24151078, 22238380, 21149580, 18842585, 16150737 |
| DNA2 | - | GO: G4 DNA unwinding | 5 | 3 | -2.2 | 0 | - | 23604072 |
| FUS | YES | - | 3 | 2 | -3.2 | - | - | 23521792 |
| MCRS1 | - | GO: G4 RNA binding | 2 | 0 | - | 1 | -1.7 | 16571602 |
| RECQL4 | YES | - | 6 | 3 | -1.8 | 2 | -2.6 | 25336622 |
| SF3B3 | YES | - | 4 | 0 | - | 3 | -2.9 | 23381195 |
| XRN1 | - | GO: G4 DNA binding, G4 RNA binding | 4 | 3 | -1.1 | 3 | -2.7 | 9049243 |

**F** Manually curated top ranked G4-related genes (from Pubmed/PMC)

| Gene | PolySearch Score | Hairpins in t0 | Depleted hairpins (PDS) | Median log2FC (PDS) | Depleted hairpins (PhenDC3) | Median log2FC (PhenDC3) | PUBMED ID | G4 association |
|------|------|------|------|------|------|------|------|------|
| DHX36 | 855 | 5 | 1 | -1.9 | 4 | -4.6 | 29269411, 28069994, 25653156, 25611385, 24151078, 22238380, 21149580, 18842585, 16150737 | DNA and RNA G4 helicase |
| ATRX | 205 | 16 | 1 | -1.6 | 5 | -1.4 | 23563309 | Chromatin remodeller and DNA G4 binder |
| RECQL4 | 130 | 6 | 3 | -1.8 | 2 | -2.6 | 25336622 | Cooperates with BLM to unwind DNA G4 |
| CNBP | 115 | 5 | 1 | -2.2 | 3 | -5.5 | 23774591, 28329689,24594223, 26332732 | Promotes formation of G4 at Myc promoter (NHE III region) |
| RTEL1 | 75 | 4 | 1 | -3.0 | 2 | -4.8 | 22579284, 24115439, 19596237 | DNA G4 helicase at telomeres |
| DNA2 | 55 | 5 | 3 | -2.2 | 0 | - | 18593712, 23604072 | DNA G4 helicase/nuclease at telomeres |
| TFAM | 55 | 5 | 2 | -1.2 | 5 | -3.0 | 28276514 | DNA/RNA G4 binder in mitochondrial DNA |
| TOP1 | 45 | 8 | 3 | -3.1 | 4 | -3.9 | 11756434, 25473964 | Prevents G4 formation |
| PCNA | 35 | 6 | 3 | -5.6 | 4 | -2.6 | 24115439 | Required for unwinding of telomeric G4 by RTEL1 |
| PPM1D | 35 | 6 | 5 | -1.8 | 1 | -1.9 | 23514618 | DDR, response to G4 ligand |
| RAD17 | 35 | 5 | 4 | -1.9 | 2 | -2.6 | 17932567 | DDR, response to G4 ligand |
| CHEK1 | 30 | 14 | 4 | -3.2 | 7 | -2.7 | 24441772, 23396447, 27597417, 24464582, 26845351 | DDR, response to G4 ligand |
| EEF1A1 | 15 | 1 | 0 | - | 1 | -5.1 | 27736771 | Proteomic protein binder to RNA G4 |
| ORC1 | 15 | 7 | 1 | -2.7 | 4 | -2.7 | 24003239 | Contains G-rich RNA binding domain, proposed to preferentially bind G4 |
| RPA3 | 15 | 13 | 3 | -1.9 | 5 | -3.1 | 16973897, 21772995, 24747047, 23708363, 27440048 | RPA subunit: involved in disasocciation of G4 |
| POLR1A | 10 | 4 | 2 | -3.3 | 1 | -5.1 | 27391441 | G4 ligand treatment inhibits RNA pol 1 |

**Figure 3.** Genome-wide screening in A375 cells reveals deficiencies in known G4-associated genes as sensitive to G4-stabilising small molecules. (A–C) Venn diagrams for: (A) significantly differentially expressed individual shRNAs (FDR ≤ 0.05); (B) significantly depleted genes (50% or three hairpins, FDR ≤ 0.05, median log2FC < 0) following DMSO, PDS and PhenDC3 treatment and (C) Significant PDS and PhenDC3 sensitiser genes not in DMSO and after applying a median log2FC ≤ −1 cut off. (D–F) Tables showing the number of depleted hairpins and median log2FC values for: (D) known G4

*Figure 3 continued on next page*

*Figure 3 continued*

ligand sensitisers, *ATRX, HERC2, BRCA1* and *BRCA2*, that are independently validated in our screen; (E) sensitisers annotated with a G4-associated term in GO, UniprotKB or G4IPBD databases and (F) sensitisers identified as G4-related by text-mining showing the associated PolySearch2 algorithm score and summary of the G4 association. Sensitisers are defined as a gene where 50% or three hairpins were significantly differentially expressed (FDR $\leq$ 0.05) with median $\log_2$FC $\leq$ −1. See also *Supplementary file 1*.

DOI: https://doi.org/10.7554/eLife.46793.005

−1 (*Figure 3C*). It is reassuring that in this list we independently validated the known G4 sensitisers *BRCA1/2, ATRX* and *HERC2* (*McLuckie et al., 2013*; *Wang et al., 2019*; *Watson et al., 2013*; *Wu et al., 2018*; *Xu et al., 2017*; *Zimmer et al., 2016*; *Figure 3D*).

We next explored further genes already implicated in G4 biology, but whose deficiency has not yet been linked with any enhanced sensitivity to G4 ligands. For genes annotated with G4-related terms in the UniprotKB, Gene Ontology (GO) and G4IPDB databases (*Mishra et al., 2016*), an additional eight sensitisers (*ADAR, DHX36, DNA2, FUS, MCRS1, RECQL4, SF3B3* and *XRN1*) were uncovered (*Figure 3E*). Text-mining with G4 search terms using PolySearch2 on PubMed abstracts and open access full texts (see Materials and methods; *Liu et al., 2015b*) revealed a further 12 sensitisers arising from our screen including helicases (*RTEL1*), DDR components (*CHEK1, RAD17*), transcriptional proteins (*POLR1A, CNBP*) and replication factors (*ORC1, RPA3, TOP1*) (*Figure 3F*).

Within the total 758 G4-sensitiser gene list, we uncovered five significant enriched KEGG pathway clusters (p<0.05): 'cell cycle', 'ribosome', 'spliceosome', 'ubiquitin-mediated proteolysis' and 'DNA replication' (*Figure 4A*, *Supplementary file 1*). Within each cluster are gene targets common to both G4 ligands, as well as genes unique to each ligand. To gain functional insights, enriched GO 'Biological Process' and 'Molecular Function' terms were determined (*Figure 4B*; *Supplementary file 1*) which showed 20 out of 45 of the former and all the latter terms into DNA or RNA classifications, consistent with PDS/PhenDC3 directly binding nucleic acid G4 targets. Furthermore, when protein domains were considered using GENE3D and PFAM databases (*Figure 4C*), we discovered enrichments in helicase C-terminal domains, RNA recognition motifs including RRM, RBD and RNP domains, and DNA-binding domains including zinc fingers, bZIP motifs and HMG boxes. Consistent with the ubiquitin-mediated proteolysis KEGG cluster, enrichments in multifunctional ATPase domains and in ubiquitin hydrolase domains, were also found. These latter findings suggest important areas of biology not previously known to be affected by G4 intervention in mammalian cells.

## Cancer-associated gene depletion enhances sensitivity to G4-ligands

We next used the complete list of 758 genes, identified as stringent G4 ligand sensitisers above, to discover new cancer-associated gene vulnerabilities to G4-stabilising ligands. For this, we searched this list for any significant enrichment in the COSMIC database (v83) of genes causally implicated in cancer (*Forbes et al., 2015*). Of the 758 sensitisers, there was a two-fold enrichment (p=9.1×10$^{-6}$) for 50 cancer-associated genes, which increases to three-fold (p=2.5×10$^{-3}$) when considering only sensitisers common to both G4 ligands (*Figure 5A,B*, *Supplementary file 1*). Notably, when STRING network analysis (*Szklarczyk et al., 2017*) was used to investigate functional interactions, this revealed a DDR cluster that included *BRCA1* and *BRCA2*, as well as their interacting tumour suppressor partners *PALB2* and *BAP1*, two cancer-associated DDR genes not previously indicated as G4 ligand sensitisers. (*Figure 5C*). This analysis also identified as sensitisers a cluster consisting of several chromatin modifiers including *SMARCA4, SMARCB1* and *SMARCE1*.

## Focused G4-sensitiser shRNA screening reveals robust G4-ligand genetic vulnerabilities and potential therapeutic targets

To enable more rigorous and further comparative analyses that focus solely on G4 sensitisers, we developed a custom shRNA screening panel encompassing the gene sensitisers identified above plus additional G4-associated genes noted from the literature (*Figure 6A*, *Figure 6—figure supplement 1*, see Materials and methods). This panel consisted of a single retroviral shRNA pool to allow all shRNAs to be screened simultaneously under standardised conditions and to minimise technical fluctuations. We first used this panel to recapitulate the findings of the genome-wide screen above

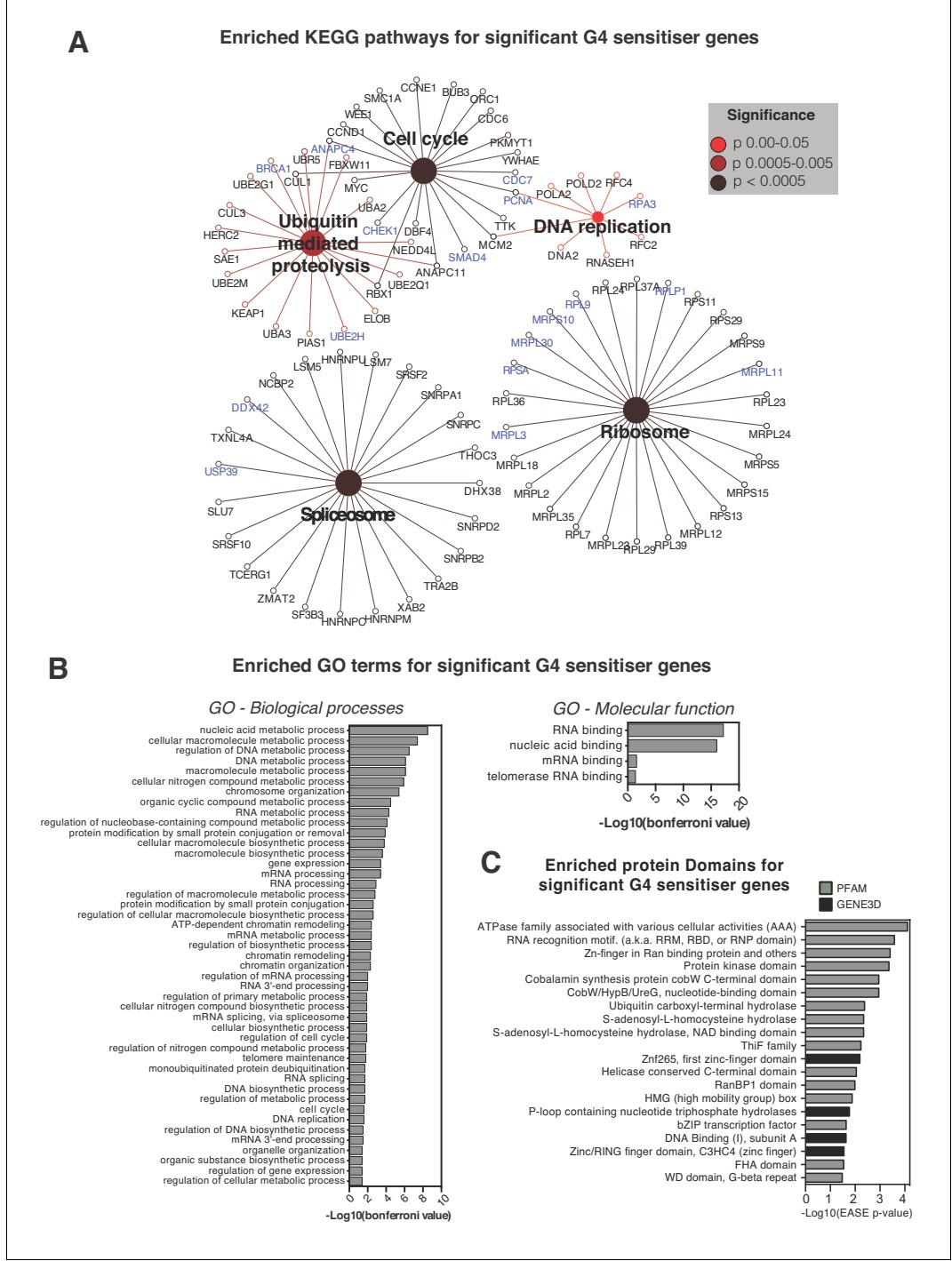

**Figure 4.** Pathways and processes showing sensitivity to G4-stabilising ligands. (**A**) Enriched KEGG pathways and (**B**) Gene Ontology terms, GO Biological Processes (BP) and Molecular Functions (MF), for the 758 genome-wide G4-sensitiser genes. Blue- genes common to both ligands; black-genes unique to either PDS or PhenDC3. A right-sided enrichment test with Bonferroni correction used (see Materials and methods). (**C**) Enriched protein domains (p≤0.05) within GENE3D (black) and PFAM databases (grey) ordered by -Log10 (EASE p-value). See also **Supplementary file 1**.
DOI: https://doi.org/10.7554/eLife.46793.006

and compare responses with different G4 ligands. Using A375 melanoma cells with PDS and PhenDC3, the custom panel recovered a total of 342 G4 sensitisers corresponding to 40.6% overlap (308 genes) with the complete genome-wide screen (**Figure 6B,C**). From this, we identified 290 G4

sensitisers with 89 and 161 unique for PDS and PhenDC3, respectively, and 40 genes common for both ligands (*Figure 6—figure supplement 1E*). Comparing PDS and PhenDC3 sensitisers by KEGG analysis shows that each ligand mostly interacts with different but related pathways (*Figure 6D,E*). Consistent with direct G4-targeting, nucleic-acid-related GO terms were enriched (*Figure 6—figure supplement 1F & G*, *Supplementary file 2*). We next considered that the 40 sensitiser genes common between PDS and PhenDC3 reflected the most robust sensitisers for G4 ligands in general and it is notable that 27 out of 40 associated with DNA or RNA binding processes, such as chromatin modification, replication transcription, and translation (*Figure 6F*). Again, the ubiquitin processes, which previously were not linked with G4 biology, were also uncovered as a significant sensitiser pathway. Overall, these results clearly show the spectrum of biological vulnerabilities that underpin the observed enhanced sensitivities for each G4-targeting ligand.

We next reasoned that the robust set of 290 G4 ligand sensitiser genes above provides a suitable test bed for exploring the arising therapeutic potential for combinatorial pharmacological inhibition and G4-ligands. We therefore looked for the presence of these sensitisers genes within the druggable genome interaction database (DGIdb) (*Griffith et al., 2013*). A total of 74 G4-sensitisers were found in the classifications 'Druggable Genome' (genes with known or predicted drug interactions) and 'Clinically Actionable' (genes used in targeted clinical cancer sequencing for precision medicine) with 13 being common to both classifications (*Figure 6G*, *Supplementary file 1*). Notably, this included KEAP1, an E3 ubiquitin ligase adapter protein and highlights a new therapeutic domain for the application of G4-based drugs. Performing a similar analysis on the 40 most robust sensitisers common to both G4 ligands gave 12 genes within DGIdb (*Figure 6H*, *Supplementary file 1*), including 5 (*BRCA1, CHEK1, CDK12, TOP1, PDKP1*) common to both druggable and clinically actionable classifications. These results therefore open up new possibilities for cancer therapies based on vulnerabilities to G4 ligands.

## G4 sensitisers common to two independent cell lines

We next sought to extend the use of the custom shRNA lentiviral library to gain initial insights into possible commonalities and differences in the response to G4 ligands in cells from different lineages. We therefore applied the custom library to mesenchymal-derived HT1080 fibrosarcoma cells (wild-type *TP53*, driven by activated *NRAS* (Q61K) and *IDH1* mutation (R132C)) and compared the results to those from ectodermal A375 melanoma cells above (*Figure 7*, *Figure 7—figure supplement 1F & G*, *Supplementary files 1 & 2*). The custom HT1080 screen recovered a total of 121 G4 ligand sensitisers, with the majority (73 genes, 58%) shared with those seen for each ligand in the A375 genome-wide screen. Cytoscape network analysis (*Figure 7A*) revealed a core set of G4-associated genes/pathways for these genes in spliceosome, HR and ubiquitin-mediated proteolysis processes (p<0.0005). Overall, 29 PDS and 22 PhenDC3 gene sensitivities were found to be shared across all three screens (*Figure 7B,C*), and it is noteworthy that both G4 ligands targeted similar processes including transcription, splicing and ubiquitin-mediated proteolysis (*Figure 7D,E*).

## *BRCA1, TOP1, DDX42* and *GAR1* are key G4 ligand sensitiser genes

When we evaluated the data collectively from all screens, it was apparent that four genes were repeatedly found as G4 ligand sensitisers- *BRCA1, TOP1, DDX42* and *GAR1*, as they consistently appeared in both cell types and with both G4-ligands in all screens (*Figure 7F*, *Figure 7—figure supplement 1F*). To corroborate these genes as genuine G4 sensitisers, we developed an independent siRNA knockdown approach using a shorter timeframe (~6 days) to recapitulate ligand-induced growth inhibition (*Figure 8*). Both A375 and HT1080 cells were transfected with siRNAs targeting *BRCA1, TOP1, DDX42* or *GAR1* alongside non-targeting siRNA and non-transfected controls. Following 24 hr, cells were treated with two concentrations of PDS and PhenDC3 or vehicle control DMSO for 144 hr. Growth curves for non-transfected and non-targeting siRNA controls were similar across ligand treatments in both cell lines (*Figure 8—figure supplements 1* and *2*). For both HT1080 (*Figure 8A & B*) and A375 cells (*Figure 8—figure supplement 3A & B*), protein depletion following siRNA transfection was confirmed after 48 hr by immunoblotting cell lysates with the appropriate antibodies (average 76–92% knockdown for HT1080; 41–69% knockdown for A375 after 48 hr). The percentage difference in confluency compared to non-targeting siRNA control cells was

**A**  PDS sensitisers

| Gene | total hairpins in t0 | significant sensitizer hairpins in tF PDS | median logFC PDS |
|---|---|---|---|
| DDX10 | 8 | 3 | -3.9 |
| FIP1L1 | 4 | 2 | -3.7 |
| FUS | 3 | 2 | -3.2 |
| TOP1 | 8 | 3 | -3.1 |
| CCND1 | 10 | 3 | -3.1 |
| BAP1 | 8 | 4 | -3.1 |
| MYC | 8 | 6 | -3.0 |
| SPEN | 4 | 3 | -3.0 |
| CCDC6 | 4 | 2 | -2.9 |
| BRCA1 | 5 | 3 | -2.8 |
| SMARCE1 | 16 | 9 | -2.4 |
| CLTC | 5 | 3 | -2.3 |
| CBFB | 4 | 2 | -2.2 |
| ARNT | 5 | 3 | -2.2 |
| BRCA2 | 3 | 3 | -2.2 |
| SMAD4 | 4 | 2 | -2.1 |
| SMARCB1 | 8 | 4 | -2.0 |
| WHSC1 | 16 | 5 | -1.9 |
| RECQL4 | 6 | 3 | -1.8 |
| PPM1D | 6 | 5 | -1.8 |
| SMARCA4 | 14 | 5 | -1.8 |
| CDK12 | 10 | 4 | -1.7 |
| NKX2-1 | 2 | 1 | -1.6 |
| DEK | 7 | 3 | -1.4 |
| PAX7 | 4 | 2 | -1.4 |
| RAF1 | 7 | 3 | -1.3 |
| PALB2 | 5 | 3 | -1.2 |
| ZRSR2 | 8 | 4 | -1.2 |

**B**  PhenDC3 sensitisers

| Gene | total hairpins in t0 | significant sensitizer hairpins in tF PhenDC3 | median logFC PhenDC3 |
|---|---|---|---|
| CNBP | 5 | 3 | -5.5 |
| ALK | 10 | 3 | -4.7 |
| TOP1 | 8 | 4 | -3.9 |
| ARNT | 5 | 4 | -3.7 |
| FOXL2 | 3 | 2 | -3.4 |
| KEAP1 | 4 | 3 | -3.3 |
| RANBP2 | 11 | 3 | -3.2 |
| EWSR1 | 5 | 3 | -3.2 |
| FIP1L1 | 4 | 2 | -3.0 |
| DDX6 | 5 | 3 | -2.9 |
| CDK12 | 10 | 4 | -2.8 |
| FGFR3 | 10 | 3 | -2.7 |
| HSP90AB1 | 8 | 4 | -2.4 |
| CCDC6 | 4 | 3 | -2.4 |
| PPP6C | 6 | 3 | -2.4 |
| SMAD4 | 4 | 3 | -2.3 |
| PLCG1 | 12 | 3 | -2.2 |
| BRCA1 | 5 | 3 | -2.1 |
| CARS | 6 | 3 | -2.0 |
| CREB3L1 | 4 | 2 | -1.9 |
| WHSC1 | 16 | 4 | -1.9 |
| UBR5 | 3 | 3 | -1.8 |
| BRCA2 | 3 | 2 | -1.8 |
| SRSF2 | 7 | 3 | -1.8 |
| YWHAE | 6 | 3 | -1.5 |
| NBN | 6 | 3 | -1.4 |
| ATRX | 16 | 5 | -1.4 |
| RUNX1 | 4 | 2 | -1.4 |
| DEK | 7 | 4 | -1.4 |
| CCNE1 | 8 | 3 | -1.3 |
| NF2 | 4 | 2 | -1.3 |
| CYLD | 8 | 3 | -1.2 |

**C**

**Figure 5.** Identification of cancer-associated genes whose loss promotes sensitivity to G4 ligands. (A, B) Median $\log_2$FC and number of significantly depleted hairpins for G4 sensitisers overlapping the COSMIC database for PDS (A) and PhenDC3 (B). Genes common to both are indicated in blue. See also *Supplementary file 1*. (C) Functional interaction network analysis using STRING for the 50 COSMIC proteins indicated in A and B. Clusters are shown using confidence interactions > 0.4 from co-expression and experimental data. Box indicates the DDR cluster.

*Figure 5 continued on next page*

*Figure 5 continued*

DOI: https://doi.org/10.7554/eLife.46793.007

plotted (*Figure 8—figure supplement 1B–E* and *Figure 8—figure supplement 3C–F*) and compared to DMSO treatment at 72, 96 and 120 hr (*Figure 8C–F*, *Figure 8—figure supplement 3G–J*).

Mirroring the shRNA screen findings, siRNA knockdown of all four genes in HT1080 cells imparted significant increases in sensitivity with PDS or PhenDC3 compared to DMSO. Some differences between the ligands and individual gene knockdowns were noted. For *BRCA1* and *TOP1* the lowest concentration of PDS resulted in the most sensitisation and this was evident early at 72 hr, whereas both PhenDC3 concentrations resulted in similar growth inhibition and was apparent later (*Figure 8C & D*, *Figure 8—figure supplement 1*). For DDX42 and GAR1, growth inhibition was mostly manifest from 96 hr, with both ligands and concentrations being broadly similar (*Figure 8E & F*, *Figure 8—figure supplement 1*). Results with the A375 cells also lend support to our observations, although there were some differences compared to HT1080 cells (*Figure 8—figure supplements 2* and *3*). While *GAR1* knockdown showed a similar sensitivity profile, *BRCA1* and *TOP1* deficiencies were sensitive to PDS but not PhenDC3. *DDX42* knockdown in A375 cells did not reflect the screens ligand sensitivities and this may in part be due to lower knockdown efficiency compared (~40%). Nonetheless, these independent siRNA short-term assays substantiate that *BRCA1, TOP1, DDX42* and *GAR1* are genetic vulnerabilities to G4 ligands and these may open up future possibilities for therapeutic development.

## G4-targeting ligands plus pharmacological inhibitors of G4 sensitiser genes demonstrate synergistic cell killing

One of our aims was to identify potential cancer genotypes where G4-ligands could be therapeutically exploited. Cancers deficient in our newly discovered G4 sensitisers may be preferentially sensitive to G4-ligands as single agents. Alternatively, rather than exploiting a genetic deficiency per se, it may be possible to use pharmacological inhibition of a critical cancer gene product that phenocopies the deficiency in combination with G4 ligands as an orthogonal approach (*Figure 9A*). As proof-of-principle, we systematically evaluated cell death potentiation with the G4 ligand PDS in combination with pharmacological inhibitors for two new G4 sensitisers gene products, the WEE1 kinase or the deubiquitinase USP1 (*Figure 9B*). WEE1 is a crucial G2/M regulator overexpressed in several cancers (*Matheson et al., 2016*), and USP1 is involved in DDR regulation and is overexpressed in non-small cell lung and other cancers (reviewed in *García-Santisteban et al., 2013*). For our studies, we used MK1775 (AZD1775), a WEE1 kinase inhibitor that is being clinically evaluated in several cancers (*Richer et al., 2017*), and pimozide a potent USP1-targeting drug (*Chen et al., 2011a*). HT1080 and A375 cells were cultured in matrix combinations of PDS with MK1775 or pimozide at concentrations surrounding the $GI_{50}$ values and cell viability measured after 96 hr using an end-point ATP luminescence-based assay (CellTiter-Glo, Promega). Combenefit software (*Di Veroli et al., 2016*) was then used to calculate synergy for different treatment combinations in which the percentage growth inhibition compared to single agent controls is used to plot a 3D-dose-response surface of synergy distribution in concentration space (*Figure 9C–F*). In HT1080 cells, synergy was found for both PDS and MK1775 or pimozide combinations (*Figure 9C,D*, *Figure 9—figure supplement 1*) with peak synergies of 21% and 24% at 156 nM PDS with 21 nM MK1775 or 6.25 µM pimozide, respectively ($GI_{50}$ for PDS, MK1775 and pimozide alone = 322 nM, 59 nM and 8.4 µM, respectively). A375 cells showed lower synergy with PDS and MK1775 combination (*Figure 9E*, *Figure 9—figure supplement 1*), with peak synergy of 15% at 8 µM PDS, 444 nM MK1775 ($GI_{50}$ for PDS, MK1775 and pimozide alone = 8.5 µM, 625 nM and 12.2 µM, respectively). The greatest synergy was seen in combinations of PDS and pimozide in A375 cells (*Figure 9F*, *Figure 9—figure supplement 1*) with a peak synergy of 61% at 5.33 µM PDS, 6.25 µM pimozide. Furthermore, long-term clonogenic survival assays revealed a similar potentiation of growth inhibition, albeit at lower compound concentrations, for PDS/MK1775 and PDS/pimozide drug combinations for both cell lines tested (*Figure 9—figure supplement 2*). Altogether, these results validate that appropriate drug combinations can synergistical act as a surrogate for gene deficiencies in the presence of G4 ligands and thus complements the findings uncovered by our genetic screening approach.

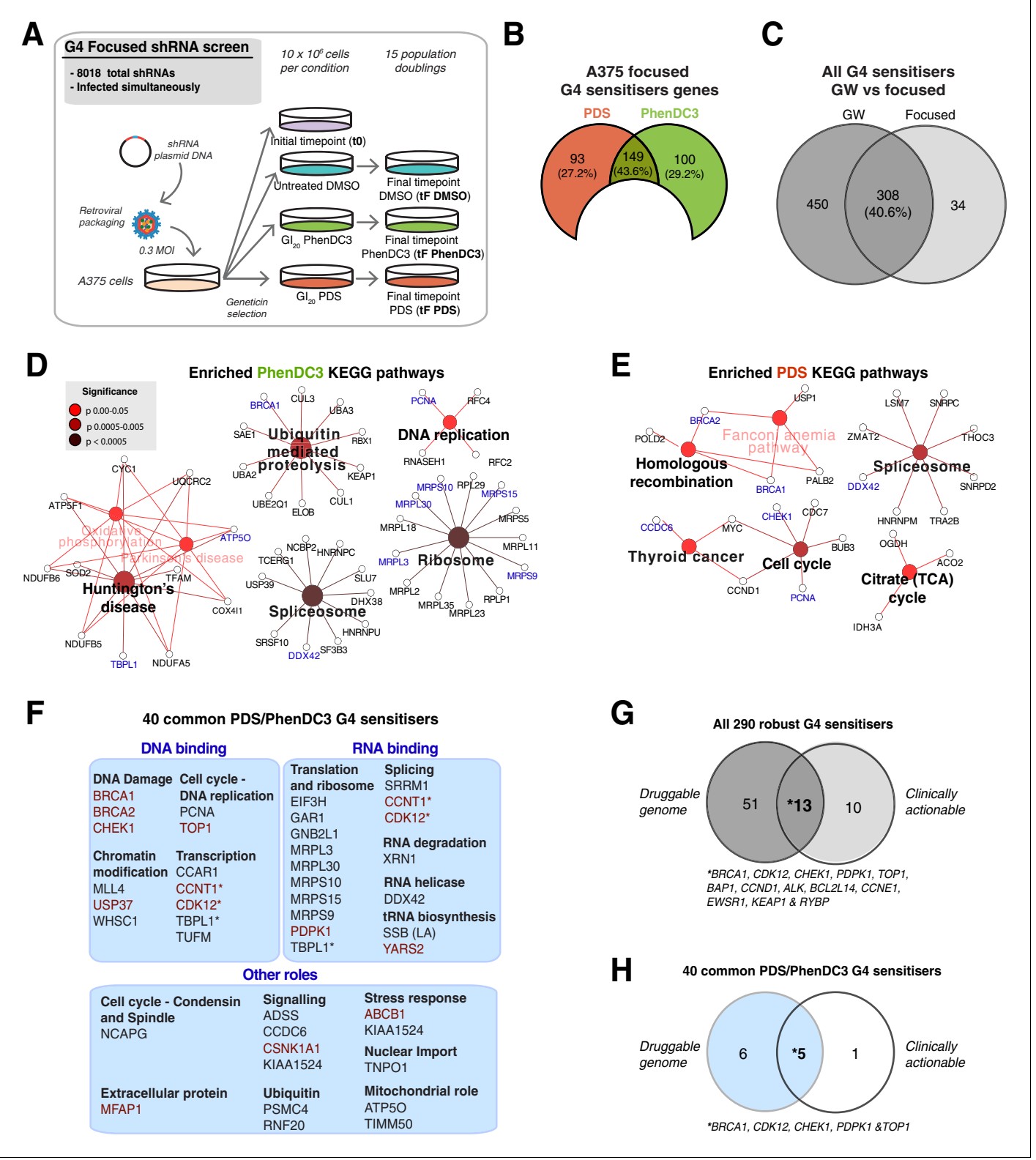

**Figure 6.** A custom G4 sensitiser shRNA panel reveals unique and common G4 ligand sensitivities. (**A**) A shRNAs custom retroviral pool (~8000 hairpins) was used to infect A375 cells. Following antibiotic selection, the reference time point (t0) was taken and then cells were cultured for 15 population doublings in DMSO, PDS or PhenDC3 before (tF). Three biological replicates were performed. (**B**) Significant sensitiser genes for the A375 focused screen (50% or three significantly depleted with median $\log_2$ FC$\leq -1$). (**C**) Overlap of the genome-wide (GW) with A375 focused screen for PDS and PhenDC3 G4-sensitisers combined (see also **Figure 6—figure supplement 1**). (**D–E**) Enriched KEGG pathways for (**D**) PhenDC3 and (**E**) PDS sensitiser

*Figure 6 continued on next page*

*Figure 6 continued*

genes common to the genome-wide and A375 focused screens. A right-sided enrichment test with Bonferroni correction used (see Materials and methods). (F) DAVID, STRING (experimental data, co-expression, medium confidence ≥0.4) interaction and UniprotKB data were used to categorise biochemical roles for the 40 high-confidence G4 sensitisers common to both ligands. Genes in red indicate those found in the (DGIdb 2.0). *=genes in multiple categories. (G, H) Overlap of the all 290 robust G4 sensitisers (G) and the 40 G4 sensitisers common to both ligands (H) with the Drug Genome Interaction database. The druggable genome denotes genes with known or predicted drug interactions. Clinically actionable denotes genes used in targeted cancer clinical sequencing panels. See also *Figure 6—figure supplement 1*, *Supplementary file 1*.

DOI: https://doi.org/10.7554/eLife.46793.008

The following figure supplement is available for figure 6:

**Figure supplement 1.** Focused A375 cell screening parallels findings from the genome-wide screen and highlights differences in individual G4 ligand sensitivities.

DOI: https://doi.org/10.7554/eLife.46793.009

## Identification of DDX42 as a new G4-binding protein

Another of our aims was to use the findings of our shRNA screen to identify proteins that may bind and/or regulate G4 structures in cells, such as G4 helicases. Indeed, DHX36 and DHX9, known G4 helicases (*Giri et al., 2011*; *Chen et al., 2018*; *Chakraborty and Grosse, 2011*; *Creacy et al., 2008* ; *Vaughn et al., 2005*) and the DEAD box protein DDX3X, that was recently shown to bind RNA G4s (*Herdy et al., 2018*), were identified as G4 sensitisers in our screen. Further members of the DDX/DHX helicase family also appeared as G4 sensitisers (*Figure 10A*), raising the question of whether these represent previously uncharacterized G4-binding proteins. To address this directly, we chose to investigate DDX42 as this was one of the four key G4 sensitisers identified above. DDX42 is a non-processive RNA helicase (*Uhlmann-Schiffler et al., 2006*) and has been associated with splicing (*Will et al., 2002*); however, this protein remains largely uncharacterised. By immuno-blotting of nuclear and cytoplasmic sub-cellular fractions (*Figure 10B–E*), we first established that DDX42 predominantly localises to the nucleus (~4 to 9-fold greater than cytoplasmic levels) in three independent cell lines, (HT1080, HEK293 and HeLa). As controls for fractionation, LaminB1 and GAPDH were found to partition as expected into nuclear and cytoplasmic fractions, respectively (*Figure 10C,D*).

As DDX42 is known to bind RNA, we next set out to demonstrate DDX42 affinity for a RNA-G4 structure as this has not previously been documented. For this, a G4 RNA oligonucleotide from the NRAS 5'UTR sequence, which forms a stable parallel G4 (*Kumari et al., 2007*), was used together with a mutated oligonucleotide unable to form a G4 structure and also a RNA hairpin as negative controls (*Herdy et al., 2018*). Oligonucleotides were folded in 100 mM KCl to promote G4 structure formation and the resultant structures confirmed by circular dichroism (CD) spectroscopy (*Figure 10—figure supplement 1*). The affinity of recombinant DDX42 was then investigated by Enzyme Linked Immunosorbent Assay (ELISA, *Figure 10F*) and binding parameters calculated using a non-linear regression model, assuming one-site-specific binding and saturation kinetics using Prism software. DDX42 bound the NRAS G4 folded in KCl with an apparent $K_d$ of 71.1 ± 3.5 nM and did not bind detectably to the mutant oligonucleotide or RNA hairpin controls.

Given the nuclear localisation of DDX42 and as some DDX proteins also have DNA helicase activity (*Kikuma et al., 2004*), the DDX42 affinity for a DNA G4 structure was investigated. For this, an oligonucleotide corresponding to the stable parallel G4 structure in the promoter of *MYC* (*González and Hurley, 2010*; *Yang and Hurley, 2006*), and a non-G4 forming control, were used. The oligonucleotides were folded in 100 mM KCl and structures verified by CD spectroscopic analysis (*Figure 10—figure supplement 1B*). DDX42 affinity by ELISA (*Figure 10G*) showed that DDX42 binds to the MYC DNA G4 with an apparent $K_d$ of 232.9 ± 23.5 nM with little binding to the mutant control. Thus, the G4 sensitiser screen has enabled us to identify and classify DDX42 as a G4-interacting protein as a new finding.

## Discussion

G4 structures are emerging as promising clinical targets in cancer (*Xu et al., 2017*) but the range of disease-associated genetic backgrounds that potentiate G4 ligand effects has yet to be defined. Here, we have discovered many genes that when depleted enhance cell killing with the G4 ligands

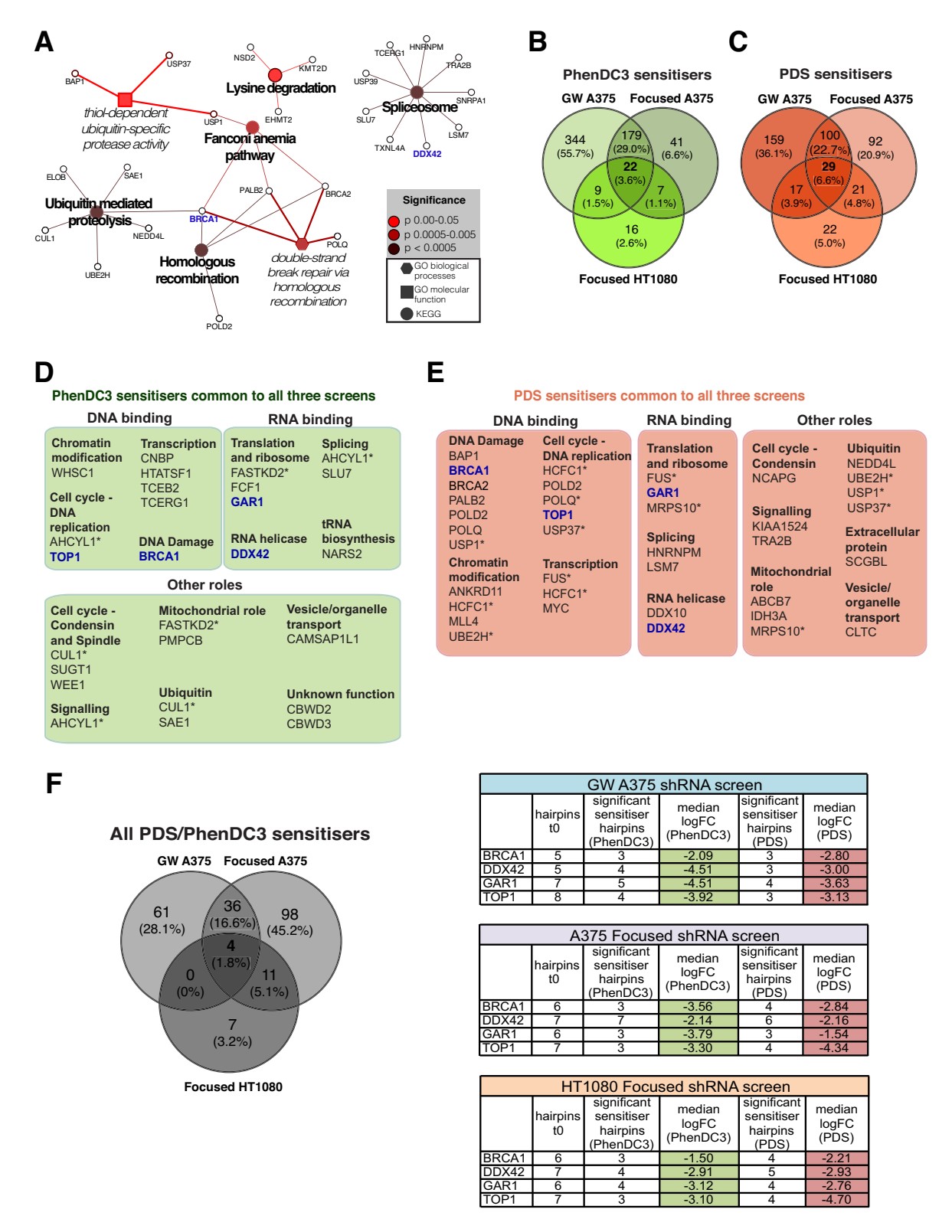

**Figure 7.** G4 sensitivities in two different cell lines. (**A**) Enriched KEGG and GO pathways for all G4 ligand-specific sensitisers (73 genes) shared between the genome-wide A375 and HT1080 screens. A right-sided enrichment test with Bonferroni correction used (see Materials and methods). (**B–C**) Comparison of G4 sensitisers across A375 focused, A375 genome-wide and HT1080 focused screens for (**B**) PhenDC3 and (**C**) PDS. (**D–E**) DAVID, STRING (experimental data, co-expression, medium confidence (≥0.4) interaction) and UniProtKB data analysis showing biochemical functions for

*Figure 7 continued on next page*

*Figure 7 continued*

common PhenDC3 (**D**) and PDS (**E**) sensitisers across all three screens. \*=genes in multiple categories. Blue, four sensitisers common to both ligands. (**F**) Left, common sensitiser genes across all three screens. Right, number of depleted hairpins and median log$_2$FC values for four key genes found as both PDS and PhenDC3 sensitisers across all the three screens. See also *Figure 7—figure supplement 1*, *Supplementary file 1*.

DOI: https://doi.org/10.7554/eLife.46793.010

The following figure supplement is available for figure 7:

**Figure supplement 1.** Focused HT1080 and A375 cell screening reveals shared PDS and PhenDC3 sensitivities.

DOI: https://doi.org/10.7554/eLife.46793.011

PDS and/or PhenDC3. The majority of these have no documented link to G4 biology and the use of low ligand concentrations is likely to favour discovery of gene losses that are the most sensitive in imparting selective cell killing. Validating the success of our approach, we independently identified G4-associated protein coding genes known to be genetic vulnerabilities to G4 ligands including *BRCA1/2, HERC2* and *ATRX* (*McLuckie et al., 2013*; *Wang et al., 2019*; *Watson et al., 2013*; *Wu et al., 2018*; *Xu et al., 2017*; *Zimmer et al., 2016*). We now report for the first time genetic vulnerabilities in 20 other known G4-associated genes that promote sensitivity to G4-stabilising ligands. These include direct nucleic acid binders and/or unwinders, such as ADAR, DHX36, DNA2, FUS, MCRS1, RECQL4, SF3B3 and XRN1.

The clinical PARP inhibitor, olaparib has exemplified the concept of synthetic lethality in BRCA-deficient cells (*Bryant et al., 2005*; *Farmer et al., 2005*), and it is notable that BRCA deficiencies were isolated as one of the top genetic vulnerabilities for both G4 ligands in both A375 and HT1080 cells. While PDS and PhenDC3 have not been optimised by medicinal chemistry, the findings of Zimmer et al showing similar efficacy of PDS and olaparib in several BRCA-deficient models (*Zimmer et al., 2016*) lends further support that our screen detects robust, biologically relevant effects.

In dropout screens, dissociating minor from robust growth effects is important and is highly dependent on parameters such as compound dose, genotype and cell line selected. Our screen was designed with stringent parameters to detect genes deficiencies worthy of further exploration. Indeed, we demonstrate potent growth inhibition of up to 80% of the four top G4 sensitisers genes in a parallel siRNA approach.

The gene sensitivities uncovered here have potential to be exploited chemotherapeutically in cancer by deploying a G4-stabilising drug as a single-agent therapy. Alternatively, in the absence of a particular gene deficiency, pharmacological inhibition of a critical oncogene could phenocopy the genetic sensitivities described here and be used in combinatorial treatments with G4-stabilising drugs. This may be attractive as cells are less likely to simultaneously develop resistance against two drugs (reviewed in *Chan and Giaccia, 2011*). Furthermore, as lower drug doses are used, this increases the therapeutic window and has less adverse side effects. As proof-as-principle for this, we selected the WEE1 cell cycle kinase and the deubiquitinase USP1, and demonstrated that their pharmacological inhibition, with MK1775 and pimozide, respectively, leads to the potentiation of cell death in conjunction with the G4 ligand PDS. For example, 5.3 µM PDS or 6.25 µM pimozide alone impart little growth inhibition (14% and 6% respectively), but together they lead to strong growth inhibition (79%). *Table 1* highlights further potential combinatorial opportunities for cancer-associated genes with clinical and/or experimental drugs. Additional therapeutic possibilities for other gene sensitivities that are largely still to be explored from a pharmacological perspective are illustrated in *Table 2*.

While the custom HT1080 screen recovered 58% of sensitisers seen for each ligand in the A375 genome-wide screen, it is striking that this increases to 93% (i.e. 112 out of 121) when considering all screens irrespective of G4 ligand, suggesting remarkable consistency when comparing G4 ligand effects globally. Differences in individual ligand sensitives may arise from variances in cellular uptake and dose, for example, the GI20 dose of PhenDC3 is ten-fold higher for A375 compared to HT1080; G4 ligand-dependent molecular preference for G-tetrad end binding (*Le et al., 2015*) and/or the accessibility of G4s in the chromatin of individual cell lines (*Hänsel-Hertsch et al., 2016*). These points plus differences in protein knockdown efficiency, especially in A375 cells, may contribute to the differences in G4 ligand growth inhibition in our siRNA experiments. In the siRNA experiments, the G4 ligand-induced growth inhibition of both A375 and HT1080 appear not to follow a 'typical'

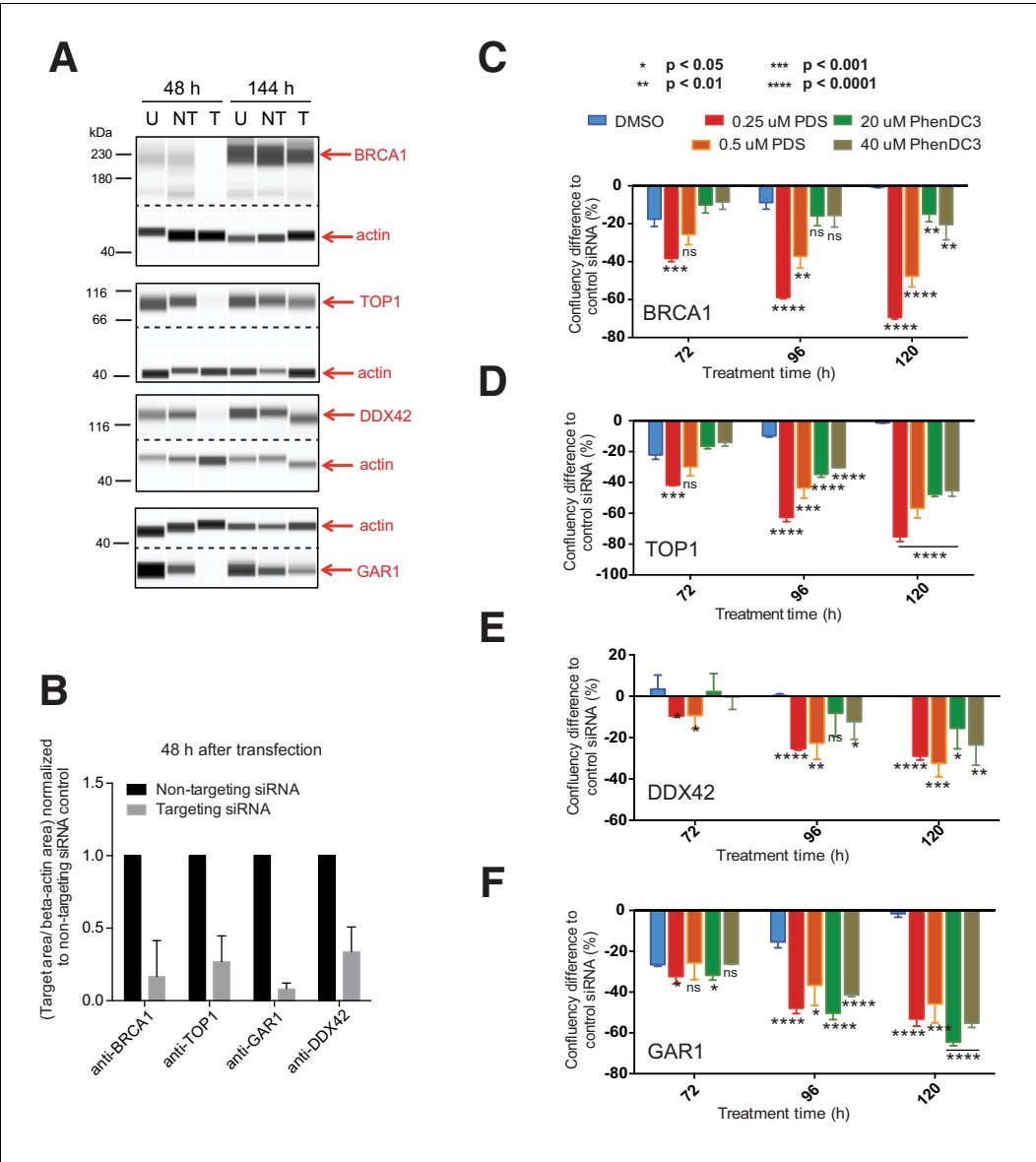

**Figure 8.** siRNA knockdowns validate *BRCA1*, *TOP1*, *DDX42* and *GAR1 as* key G4 ligand sensitiser genes. (**A**) HT1080 cells were treated with non-targeting (NT) or targeting (T) siRNAs for *BRCA1*, *TOP1*, *DDX42* and *GAR1*. 48 hr and 144 hr after transfection, cell lysates and a non-transfected cell lysate (U) were probed with appropriate antibodies and actin control by western blotting. (**B**) Protein levels for targeting (T) and non-targeting (NT) 48 hr lysates were normalised to the internal actin control and then normalised to NT levels for three biological replicates (mean ± standard deviation). (**C–F**) HT1080 cells were transfected with targeting siRNAs for 24 hr before PDS, PhenDC3 or DMSO treatment. Comparative box plots of confluency differences and significance (unpaired parametric t-test) at selected timepoints for (**C**) *BRCA1*, (**D**) *TOP1*, (**E**) *DDX42*, (**F**) *GAR* (ns = not significant) for three separate siRNA transfections. See also *Figure 8—figure supplements 1*, *2* and *3*.

DOI: https://doi.org/10.7554/eLife.46793.012

The following source data and figure supplements are available for figure 8:

**Source data 1.** Source files for western blots.
DOI: https://doi.org/10.7554/eLife.46793.017

**Figure supplement 1.** Short-term siRNA knockdowns of four key sensitisers in HT1080 cells show dose-dependent growth inhibition with G4-ligands.

DOI: https://doi.org/10.7554/eLife.46793.013

**Figure supplement 2.** Short-term siRNA knockdowns of four key G4-sensitisers in A375 cells show dose-dependent growth inhibition with G4-ligands.

*Figure 8 continued on next page*

*Figure 8 continued*

DOI: https://doi.org/10.7554/eLife.46793.014

**Figure supplement 3.** Short-term siRNA knockdowns validate four key sensitivities from the shRNA screening in A375 cells.

DOI: https://doi.org/10.7554/eLife.46793.015

**Figure supplement 3—source data 1.** Source files for western blots.

DOI: https://doi.org/10.7554/eLife.46793.016

dose response where higher concentrations lead to greater effects. This may in part be due to there being an optimum G4 ligand dose for a particular gene loss leading to enhanced cell death. Indeed, it is thought that lower drug concentrations better fall within a 'synthetic lethality window' (*Nijman, 2011*). Higher doses may mask these effects, by targeting more G4s that are not dependent on the particular gene lost and/or be due to other off-target effects. This is also supported by the experiments in *Figure 9* that show synergy is only apparent at defined concentrations.

Our data additionally provides insights into the possible functions of the identified G4 sensitisers and indicates roles in DNA damage response (DDR), transcription/chromatin remodelling, nucleic acid unwinding, splicing and ubiquitin-mediated proteolysis. Our findings substantially advance our knowledge of G4 interactions with DDR beyond *BRCA1/2* as several key HR genes were identified as novel G4-sensitisers including *PALB2*, *BAP1* and the deubiquitinase *USP1*. Importantly, this highlights that such HR repair mechanisms are an integral and important cellular response in preventing cell death induced through the increased persistence of G4s. Persistent G4 structures are also inhibitory to DNA replication/cell cycle progression (reviewed in *Valton and Prioleau, 2016* ), and it is of note that we also uncovered many cell cycle/DNA replication sensitivities such as *PCNA*, *CHEK1*, *CCND1*, *CDC7*, *RFC2* and *RFC4*. Taken together these suggest that G4 stabilisation with small molecules could be an attractive therapeutic strategy to inhibit cell growth.

Deficits in G4-unwinding helicases are predicted to increase the persistence of G4 structures resulting in heightened sensitivity to G4 ligands. Several known G4-associated helicase deficiencies were recovered, including *RECQL4*, *RTEL1* and *DHX36*, alongside many others with no known G4 link (see *Figure 10A*). Here, we demonstrate for the first time that the DDX42 DEAD/DEAH helicase is in fact a previously unidentified structure-specific G4-binding protein. On a wider level, this acts as proof-of-principle that other specific G4 interacting proteins exist within the sensitiser list of over 700 proteins. Other known G4-helicases such as BLM, WRN, PIF1 and FANCJ (reviewed in *Wu and Brosh, 2010*) were not identified as sensitisers, which may reflect functional redundancy (*Spillare et al., 2006*), or a low ligand concentrations and/or cell type effects.

Our findings highlight the ubiquitin-protesome pathway and modifications such, as neddylation as unexplored areas with respect to G4s. The only documented ubiquitin-G4 relationship in human cells is with HERC2, an E3 ubiquitin ligase that is implicated in G4 resolution whose loss sensitises cells to G4 ligands (*Wu et al., 2018*). We also independently validate *HERC2* as a G4 sensitiser in our screen and extend our observations to cover the full breadth of the proteosomal degradation pathway, including members of E1 ligase (*UBA3*, *UBA2*, *SAE1*), E2 ligase (*UBE2H*), E3 ligase (*NEDD4L*, *RBX1*, *CUL1*, *RNF20*), deubiquitinating enzyme (*USP1* and *USP37*) and proteosome (*PSMC2*) families (see *Table 2*) (*Senft et al., 2018*; *Wei and Lin, 2012*) Given the involvement of ubiquitin-proteasomal regulation in pathways, such as DDR and cell cycle, that are generally deregulated in cancer (*Harrigan et al., 2018*), this opens up an interesting intersection between ubiquitin regulation and G4s. As ubiquitin components are being targeted for anticancer therapies (*Huang and Dixit, 2016*), their efficacy might be enhanced through simultaneous G4 targeting and here we have provided strong proof-of-principle of this using synergistic combinations of pimozide (targeting UPS1) and the G4 ligand PDS.

In contrast to other genetic screens identifying sensitiser genes that enhance the efficacy of anticancer agents (*Azorsa et al., 2009*; *Martens-de Kemp et al., 2017*), our work suggests that persistent G4s are problematic for splicing. We identified several cancer-associated splicing factors as G4 sensitisers, including SRSF10, HNRNPM and the known G4-interactor FUS, which is overexpressed in several cancers (*Crozat et al., 1993*; *Dvinge et al., 2016*; *Takahama et al., 2013*). For the latter, a drug inhibiting general spliceosome assembly (*Table 1*) has been pharmacologically explored

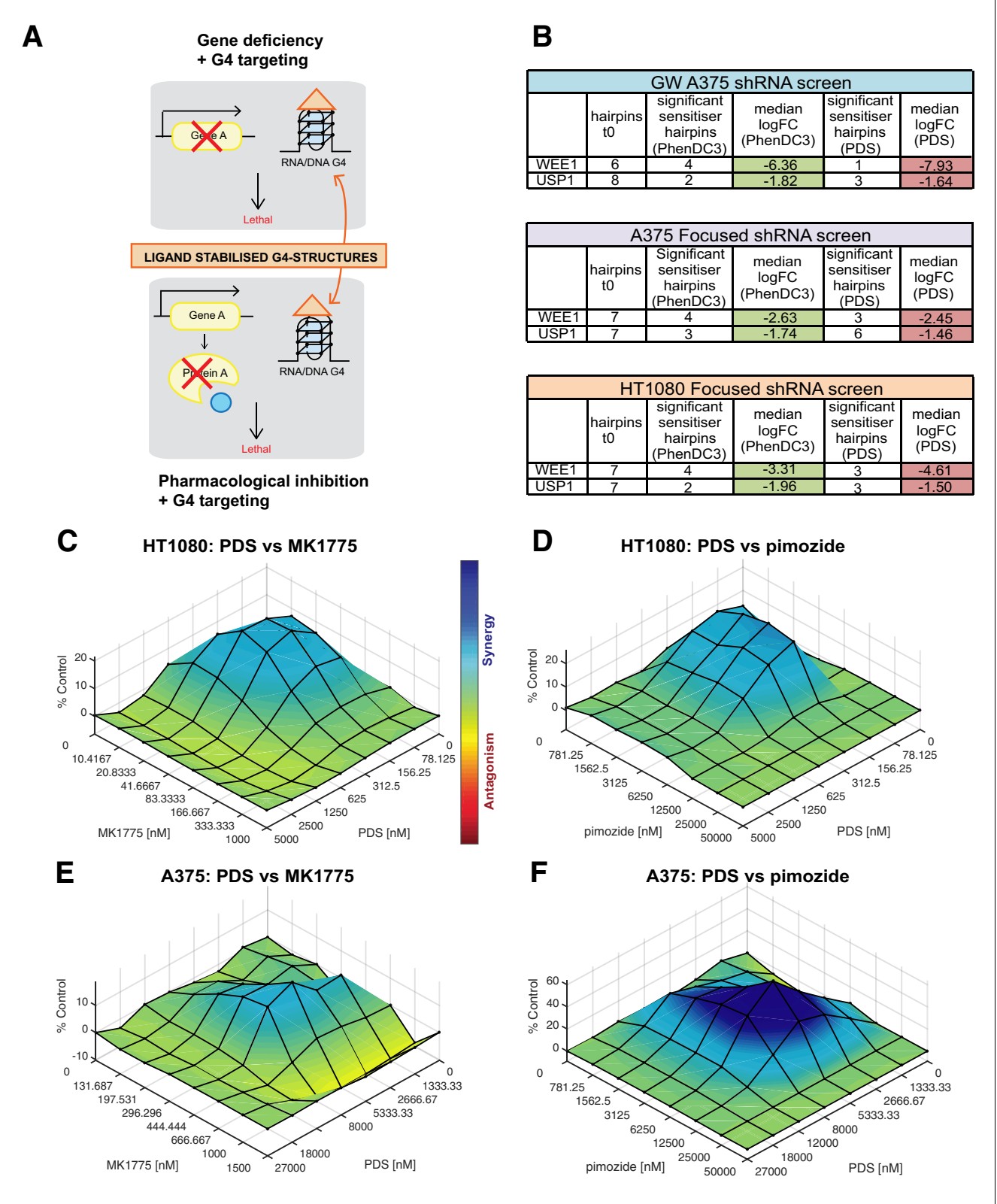

**Figure 9.** Cell death potentiation mediated by pharmacological inhibition of WEE1 or USP1 with the G4-stabilising ligand PDS. (A) Cell death potentiation with G4-stabilising ligands in combination with either gene deficiencies, such as shRNA-mediated knockdown (top), or pharmacological inhibition of a protein (bottom). (B) Numbers of depleted shRNA hairpins and median log2FC values for WEE1 and USP1 in the genome-wide and focused screens. (C–F) Synergy plots for HT1080 (C, D) and A375 (E, F) cells treated with PDS in combination with MK1775 (C, E) or pimozide (D, F). To

*Figure 9 continued on next page*

*Figure 9 continued*

determine any synergy in cell killing, 3D response surface plots were calculated using Combenefit software with the BLISS model for an average of three biological replicas. Heat bar- blue shading indicates synergy combinations, red indicates antagonism (see also *Figure 9—figure supplements 1* and *2*).

DOI: https://doi.org/10.7554/eLife.46793.018

The following figure supplements are available for figure 9:

**Figure supplement 1.** Synergy calculations for PDS with MK1775 and pimozide.
DOI: https://doi.org/10.7554/eLife.46793.019

**Figure supplement 2.** Clonogenic cell survival assay demonstrates enhanced cell death upon treatment with PDS in combination MK1775 or pimozide.
DOI: https://doi.org/10.7554/eLife.46793.020

(*Kotake et al., 2007*) raising the possibility of potentiation by G4-stabilising ligand combinatorial treatment.

We designated four of the genetic vulnerabilities as 'key' genes (*BRCA1*, *TOP1*, *DDX42*, and *GAR1)* whose deficiencies stood out with respect to consistent sensitivity to PDS and PhenDC3 in both cell lines tested. Given this, we postulate that deficiencies in any of these four genes will impart significant G4 ligand sensitivity for a range of cell types and/or with other G4 ligands. As GAR1-deficiencies are implicated in chronic lymphocytic leukaemia and contribute to telomere dysfunction (*Dos Santos et al., 2017*), we suggest that this cancer may be acutely sensitive to G4-stabilisation by small molecules.

In conclusion, we have revealed genes and pathways that interact with stabilised G4 structures. This information provides new insights into G4-related biology, especially into the functional pathways and roles as G4-interacting proteins. Furthermore, this work reveals novel disease-related genetic vulnerabilities for G4-ligands. Overall, these data provide a unique and comprehensive resource that can be further explored to understand biology that may involve G4s and also inspire new therapeutic possibilities.

# Materials and methods

**Key resources table**

| Reagent type (species) or resource | Designation | Source or reference | Identifiers | Additional information |
|---|---|---|---|---|
| Cell line (*H. Sapiens*) | A375 | ATCC | Cat# CRL-1619, RRID:CVCL_0132 | |
| Cell line (*H. Sapiens*) | HT1080 | ATCC | Cat# CCL-121, RRID:CVCL_0317 | |
| Cell line (*H. Sapiens*) | Plat-A | Cell Biolabs Inc | Cat# RV-102, RRID:CVCL_B489 | |
| Antibody | Mouse monoclonal anti-Beta Actin | Merck | Cat# A5441, RRID:AB_476744 | WB (1:250) |
| Antibody | Mouse polyclonal anti-DDX42 | Abcam | Cat# ab80975, RRID:AB_2041042 | WB (1:250) |
| Antibody | Rabbit monoclonal anti-Beta Actin | Cell Signalling Technology | Cat# 4970, RRID:AB_2223172 | WB (1:500) |
| Antibody | Rabbit polyclonal anti-BRCA1 | Cell Signalling Technology | Cat# 9010, RRID:AB_2228244 | WB (1:50) |
| Antibody | Rabbit monoclonal anti-GAPDH | Cell Signalling Technology | Cat# 5174, RRID:AB_10622025 | WB (1:50) |
| Antibody | Rabbit polyclonal anti-GAR1 | NovusBio | Cat# NBP2-31742, RRID:AB_2801566 | WB (1:100) |
| Antibody | Rabbit polyclonal anti-GST, HRP-conjugated | Abcam | Cat# ab3416, RRID:AB_30378 | ELISA (1:10,000) |

*Continued on next page*

*Continued*

| Reagent type (species) or resource | Designation | Source or reference | Identifiers | Additional information |
|---|---|---|---|---|
| Antibody | Rabbit monoclonal anti-LaminB1 | Cell Signalling Technology | Cat# 12586, RRID:AB_2650517 | WB (1:250) |
| Antibody | Rabbit monoclonal anti-TOP1 | Abcam | Cat# ab109374, RRID:AB_10861978 | WB (1:250) |
| Recombinant DNA reagent | pCMV-VSV-G plasmid | Addgene | Cat # 8454, RRID:Addgene_8454 | plasmid |
| Recombinant DNA reagent | G-quadruplex focused shRNA plasmid library | transOMIC technologies, this paper | | supplied as a glycerol stock, Materials and methods subsection: 'Composition and recombinant DNA reproduction of shRNA libraries' |
| Recombinant DNA reagent | transOMIC LMN genome-wide shRNA plasmid library | transOMIC technologies | | supplied as multiple glycerol stocks |
| Sequence-based reagent | Biotinylated oligonucleotides | *Biffi et al. (2013)*, *Herdy et al. (2018)*, this paper | | Materials and methods subsection 'Oligonucleotide annealing' |
| Sequence-based reagent | Genomic qPCR primers | this paper | | Materials and methods subsection 'Barcode recovery, adapter ligation and sequencing' |
| Sequence-based reagent | Pasha/DGCR8 siRNA | Qiagen | Cat# 1027423 | |
| Sequence-based reagent | siRNAs | this paper | | Materials and methods subsection 'siRNA validation experiments – transfection, experimental outline, immunoblotting' |
| Peptide, recombinant protein | Recombinant human DDX42 | NovusBio | Cat# H0001325-P01 | |
| Commercial assay or kit | BluePippin 2% Internal Standard Marker Kit | Sage Science | Cat# BDF2010 | |
| Commercial assay or kit | CellTitre-Glo One Solution Assay Reagent | Promega | Cat# G8461 | |
| Commercial assay or kit | KAPA library quantification kit for Illumina platforms | Kapa Biosystems | Cat# 07960140001 | |
| Commercial assay or kit | KOD Hot Start DNA polymerase | Merck | Cat# 710864 | |
| Commercial assay or kit | Lipofectamine RNAiMAX | ThermoFisher Scientific | Cat# 13778150 | |
| Commercial assay or kit | Muse Count and Viability kit | Merck | Cat# MCH600103 | |

*Continued on next page*

*Continued*

| Reagent type (species) or resource | Designation | Source or reference | Identifiers | Additional information |
|---|---|---|---|---|
| Commercial assay or kit | QIAmp DNA Blood Maxi Kit | Qiagen | Cat# 51194 | |
| Commercial assay or kit | QIAquick PCR purification kit | Qiagen | Cat# 28104 | |
| Commercial assay or kit | Qubit dsDNA HS assay kit | ThermoFisher Scientific | Cat# Q32851 | |
| Commercial assay or kit | RIPA lysis buffer | ThermoFisher Scientific | Cat# 8990 | |
| Commercial assay or kit | ZR GigaPrep Kit | Zymo Research | Cat# D4057 | |
| Chemical compound, drug | Ampicillin | Merck | Cat# A5354 | |
| Chemical compound, drug | Chloroquine diphosphate | Acros organics | Cat# 455240250 | |
| Chemical compound, drug | cOmplete mini protease inhibitor | Roche | Cat# 11836153001 | |
| Chemical compound, drug | DMSO | ThermoFisher Scientific | Cat# 20688 | |
| Chemical compound, drug | Geneticin | Gibco | Cat# 10131035 | |
| Chemical compound, drug | MK1775 | Cambridge Bioscience | Cat# CAY21266 | |
| Chemical compound, drug | PenStrep | ThermoFisher Scientific | Cat# 1507063 | |
| Chemical compound, drug | PhenDC3 | In-house synthesis | *De Cian et al., 2007a* | |
| Chemical compound, drug | Pimozide | Merck | Cat# P1793-500MG | |
| Chemical compound, drug | Pyridostatin (PDS) | In-house synthesis | *Rodriguez et al. (2008)* | |
| Chemical compound, drug | Sodium Butyrate | Merck | Cat# 303410 | |
| Chemical compound, drug | TMB substrate | Merck | Cat# T4444 | |
| Software, algorithm | Bowtie 2 v2.2.6 | *Langmead and Salzberg, 2012* | http://bowtie-bio. sourceforge.net/ bowtie2/index.shtml | |
| Software, algorithm | ClueGO v3.5.1 | *Bindea et al., 2009*; *Bindea et al., 2013* | http://www.ici.upmc.fr/ cluego/cluego Download.shtml | |
| Software, algorithm | ColonyArea | *Guzmán et al., 2014* | Image J plugin | |
| Software, algorithm | Code used for shRNA screen data analysis | This paper | All scripts are available at: https://github.com/ sblab-bioinformatics/ GWscreen_G4sensitivity | |
| Software, algorithm | Combenefit | *Di Veroli et al., 2016* | https://sourceforge. net/projects/combenefit/ | |
| Software, algorithm | Cytoscape v3.6.0 | *Shannon et al., 2003* | http://www.cytoscape.org/ | |

*Continued*

| Reagent type (species) or resource | Designation | Source or reference | Identifiers | Additional information |
|---|---|---|---|---|
| Software, algorithm | edgeR v3.6 | *Robinson et al., 2010* | http://bioconductor.org/packages/release/bioc/html/edgeR.html | |
| Software, algorithm | DAVID | *Huang et al., 2009a, Huang et al., 2009b* | https://david.ncifcrf.gov | |
| Software, algorithm | FastQC v0.11.3 | *Andrews, 2010* | http://www.bioinformatics.babraham.ac.uk/projects/fastqc | |
| Software, algorithm | FASTX-Toolkit v0.0.14 | *Gordon and Hannon, 2010* | http://hannonlab.cshl.edu/fastx_toolkit.html | |
| Software, algorithm | Graphpad Prism | GraphPad Prism (https://graphpad.com) | RRID:SCR_015807 | Version 6 |
| Software, algorithm | PolySearch2 | *Liu et al., 2015a* | http://polysearch.cs.ualberta.ca/ | |
| Software, algorithm | Python programming language v2.7.10 | https://www.python.org | | |
| Software, algorithm | R programming language v3.2.1 | https://cran.r-project.org/ | | |
| Software, algorithm | Unix tools (cat, cut, awk, sort and uniq) | https://opengroup.org/unix | | |

## Cell lines

HT1080 (RRID: CRL-1619) and A375 (RRID: CRL-121) were obtained from the American Type Culture Collection repository (ATCC) (LGC Standards, United Kingdom) and Plat-A (RRID: RV-102) was obtained from Cell Biolabs Incorporation. All cell lines were cultured in DMEM medium (Thermo-Fisher Scientific, cat #41966029) supplemented with 10% (v/v) heat inactivated FBS (ThermoFisher Scientific, cat #10500064) and grown at 37°C in a 5% $CO_2$ humidified atmosphere. Cell lines were authenticated using small tandem repeat (STR) profiling and regularly checked to be mycoplasma-free by RNA-capture ELISA. All cell lines tested negative for *Mycoplasma* contamination. None of the cell lines used in our studies was mentioned in the list of commonly misidentified cell lines maintained by the International Cell Line Authentication Committee.

## Quantification of live cell numbers

Live cell numbers (e.g. for plating cells for CellTitre-Glo assays, the screens and Incucyte experiments) were determined using the Muse Cell Analyzer (Merck), 'Count and Viability' assay according to manufacturer's instructions. Cells were diluted either 1:10 or 1:20 in 'Muse Count and Viability kit' solution (Merck, cat # MCH60013), to give a viable cell concentration of $1–2 \times 10^6$ cells/mL, with 'Events to Acquire' parameter set at 1000 events. Three cell counts were recorded.

## Determination of G4 ligand concentration for shRNA screens

PDS and PhenDC3 (both synthesised in-house) (*De Cian et al., 2007b*; *Rodriguez et al., 2008*) were used as 100 mM stocks, dissolved in DMSO (Thermofisher Scientific, cat # 20688). $GI_{20}$ values were calculated by treating A375 and HT1080 cells with serial dilutions of PDS and PhenDC3 for 96 hr and determining cell death via a CellTitre-Glo One Solution assay (Promega, cat # G8461) according to manufacturer's protocol. Each serial dilution was replicated four times for two-cell-seeding densities (1000/1500 cells per well). For both densities, curves were plotted averaging the four replicates in Prism (GraphPad v6) using a Non-Linear regression model, 'dose-response – inhibition' equation [log (inhibitor) vs. normalised response – variable slope] and $GI_{20}$ values calculated. The $GI_{20}$ concentrations used represent an average of three separate assays per cell line and yielded the following

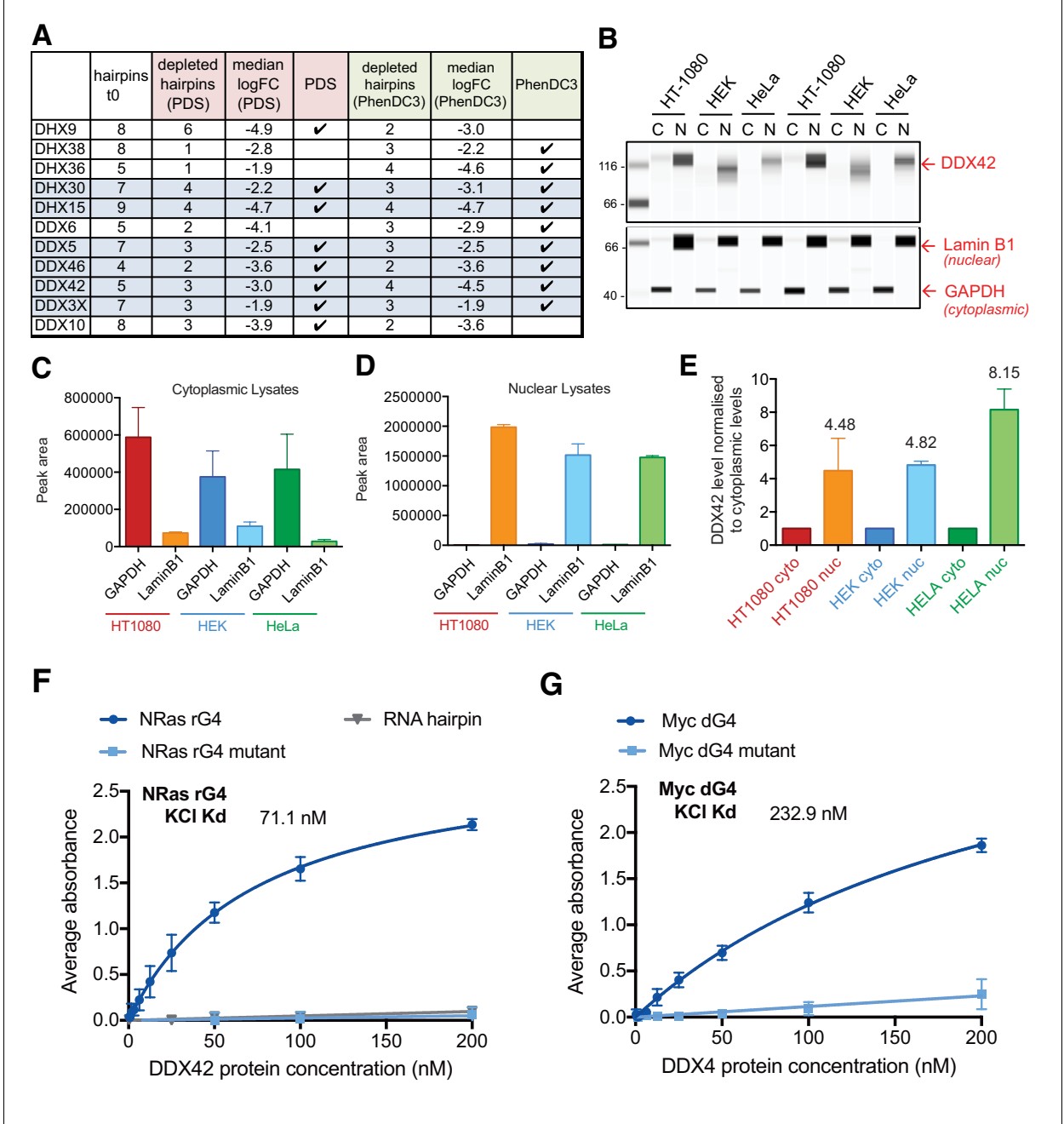

**Figure 10.** DDX42 is a predominantly nuclear G4-binding protein. (**A**) Number of depleted hairpins and median $\log_2$FC values for DEAH/DEAD-box helicase genes within the 758 genes identified in the genome-wide screen. Those highlighted in blue caused sensitivity to both PDS and PhenDC3. (**B**) Representative immunoblots showing cytoplasmic (**C**) and nuclear (**N**) lysates for HT1080, human embryonic kidney (HEK) and HeLa cells probed for DDX42, laminB1 and GAPDH2. (**C, D**) GAPDH and laminB1 protein levels for (**C**) cytoplasmic and (**D**) nuclear lysates (mean for two biological replicates ± standard deviation). (**E**) DDX42 nuclear protein levels (normalised to cytoplasmic levels, mean for two biological replicates ± standard deviation). (**F, G**) DDX42 binding curves G4s by ELISA. (**F**) NRAS 5' UTR RNA G4 (rG4), mutated G4 sequence (rG4 mut) and RNA hairpin. (**G**) MYC DNA G4 (dG4) and mutated control (dG4 mut). Apparent $K_d$ is calculated from five replicates (values are indicative as the model assumes saturation kinetics).
DOI: https://doi.org/10.7554/eLife.46793.021

The following source data and figure supplement are available for figure 10:

**Source data 1.** Source files for western blots.
DOI: https://doi.org/10.7554/eLife.46793.023

**Figure supplement 1.** Circular dichroism spectroscopy for G4 oligonucleotides.
DOI: https://doi.org/10.7554/eLife.46793.022

**Table 1.** Possible chemotherapeutic combinations for G4-stabilising ligands with clinically relevant pharmacological drugs

| Gene | Oncogene/tumour suppressor | Combinatorial/ single agent | Available drug treatments | Cancer association summary | Reference |
|---|---|---|---|---|---|
| BRCA1/ 2 | Tumour suppressor | Single agent | Olaparib CX-5461 | Deficient in ovarian, breast and colorectal cancer. | *Lee et al., 2014; Xu et al., 2017; McLuckie et al., 2013; Zimmer et al., 2016* |
| CCDC6 | Tumour suppressor | Single agent | Olaparib | Inactivated in thyroid and lung cancers. CCDC6-deficient tumours are cisplatin-resistant but olaparib sensitive. | *Puxeddu et al., 2005; Morra et al., 2015* |
| CDK12 | Oncogene | Combinatorial | Dinaclib (SCH77965) | High-grade serous ovarian cancer, often exhibits gain-of-function CDK12. | *Parry et al., 2010; Bajrami et al., 2014* |
| KEAP1 | Oncogene/Tumour suppressor | Combinatorial/ single agent | CDDO-Me CPUY192018 | KEAP1 inactivated in multiple cancers including thoracic and endometrial; also has oncogenic role, CDDO-Me used for leukaemia and sold tumours. | *Sanchez-Vega et al., 2018; Abed et al., 2015; Lu et al., 2016; Wang et al., 2014* |
| PSMC2 | Oncogene | Combinatorial | Proteosome inhibitors: Bortezomib CEP187710 Carfizomib | Ubiquitin is emerging as chemotherapeutic target, and general proteasome inhibitors clinically are used against multiple myeloma. | *Chen et al., 2011a; Mattern et al., 2012; Edelmann et al., 2011* |
| SMAD4 | Tumour suppressor | Single agent | GSKi: NCT01632306 NCT01214603 NCT01287520 | Inactivated in 50% of pancreatic adenocarcinomas. Negatively regulated by GSK, GSKis in clinical trials for metastatic pancreatic cancer and acute leukaemia. | *Schutte et al., 1996; Hahn et al., 1996; Demagny and De Robertis, 2016; McCubrey et al., 2014* |
| SRSF10 | Oncogene | Combinatorial | E7107 1C8 | Over-expressed in colon cancer. 1C8 inhibits SRSF10 and impairs HIV replication. FUS interacting protein. E7107 is a splicing inhibitor preventing spliceosome assembly. | *Zhou et al., 2014; Shkreta et al., 2017; Cheung et al., 2016; Kotake et al., 2007* |
| UBA3 | Oncogene | Combinatorial | MLN4924 | Upregulated in AML and multiple solid cancers. MLN4924 is in Phase I clinical trials. | *Soucy et al., 2009* |
| USP1 | Oncogene/Tumour suppressor | Combinatorial/ single agent | Pimozide | Over-expressed in melanoma, gastric, cervical and NSCLC; under-expressed in leukaemia and lymphoma. Pimozide is a potent USP1-targeting drug. | *García-Santisteban et al., 2013; Chen et al., 2011b* |
| WEE1 | Oncogene/Tumour suppressor | Combinatorial/ single agent | AZDMK1775 | Over-expressed in several cancers, some NSCLC are deficient. | *Matheson et al., 2016; Richer et al., 2017; Backert et al., 1999; Yoshida et al., 2004* |

*Table 1 continued on next page*

*Table 1 continued*

| Gene | Oncogene/tumour suppressor | Combinatorial/ single agent | Available drug treatments | Cancer association summary | Reference |
|------|----------------------------|-----------------------------|---------------------------|----------------------------|-----------|
| WHSC1 | Oncogene | Combinatorial | DA3003-1 PF-03882845 Chaetocin TC-LPA5-4 ABT-199 | Over-expressed in prostate cancer, multiple myeloma and mantle cell lymphoma. five potent candidate inhibitors. | *Coussens et al., 2017; Bennett et al., 2017* |

DOI: https://doi.org/10.7554/eLife.46793.024

concentrations used for the screens - A375: 10 µM PhenDC3 and 1.5 µM PDS; HT1080: 1 µM PhenDC3 and 0.5 µM PDS.

## Composition and recombinant DNA production of shRNA libraries

The genome-wide screen uses the transOMIC LMN shRNA library against the human protein coding genome, consisting of 113,002 total shRNAs, split between 12 pools for ease-of-handling (approximately 10,000 shRNAs per pool) with an average number of five optimised hairpins per gene. The G4 focused screen consists of a custom shRNA pool (transOMIC technologies) with the same LMN vector (8018 shRNAs); this includes 1247 genes (7436 shRNAs) uncovered in the genome-wide screen (751 sensitisers and 496 upregulated genes), 116 additional genes identified from the literature as potentially G4-associated (439 shRNAs) and shRNAs targeting 37 olfactory receptors as non-targeting controls (143 shRNAs). 496 upregulated genes (FDR $\leq$ 0.05, 50% or three hairpins; $\log_2 FC \geq 1$) were included to mimic the genome-wide screen on a smaller scale by maintaining the population ratio of sensitisation and resistance. In this custom pool, unlike the commercially available genome-wide library, we capped the number of shRNAs at seven per gene. The backbone of both libraries contains $Neo^R$ and ZsGreen markers to allow monitoring of infected cell lines by Geneticin (Gibco, cat # 10131035) selection and fluorescence (MacsQUANT), respectively. Both libraries were provided as glycerol stocks. Bacterial density was determined by calculating the colony-forming units (CFU) from dilutions of the original glycerol stock after plating on agar plates (overnight, 37°C, 100 µg/mL ampicillin). Glycerol stocks were thawed completely with sufficient volume taken (based on CFU) to ensure a minimum of 1000-fold hairpin representation and inoculated into liquid culture (LB media + 100 µg/mL ampicillin). Plasmid DNA was isolated using ZR Gigaprep kit D4057 (Zymo research) according to manufacturer's protocol and DNA quantified by Nanodrop One$^c$ (Thermo Fisher Scientific).

## shRNA stable cell line creation

For the genome-wide screen, each pool was treated independently, necessitating the creation of 12 different polymorphic cell lines each containing an average of 10,000 shRNAs, for both HT1080 and A375, per replica (three replicas, 36 polymorphic cell lines). Virus was produced using the Platinum-A packaging cell line (4–6 $\times$ 15 cm plates per pool) and calcium phosphate transfection. 24 hr after plating Platinum-A cells (70–80% confluency), media was replaced with DMEM medium supplemented with 1% (v/v) PenStrep (Thermo Fisher Scientific, cat # 150763) and 10% (v/v) heat inactivated FBS, shRNA library plasmid (75 µg) was then mixed with pCMV-VSV-G plasmid (7.5 µg, Addgene cat # 8454), Pasha/DGCR8 siRNA (2.7 µM, Qiagen cat # 1027423) to increase viral titre and 0.25 M $CaCl_2$ in a total volume of 1.5 mL per 15 cm dish and bubbled with 1.5 mL 2 x HBS (50 mM HEPES, 10 mM KCl, 12 mM Dextrose, 280 mM NaCl, 1.5 mM $Na_2PO_4$ at pH 7.00) and added to the Platinum-A cells (containing 17mL media) in a dropwise fashion. Immediately before adding the DNA-Pasha-transfection mixture to the Platinum-A cells, chloroquine diphosphate (lysosomal inhibitor, Acros Organics cat # 455200250) was added to the plates at a final concentration 2.5 µM. 14–16 hr after transfection, fresh media was added with 1:1000 1 M sodium butyrate (Merck, cat # 303410) for enhanced mammalian expression of the shRNA LMN plasmid. Virus was then harvested 48 hr after transfection and filter sterilised (0.45 µM) and stored at 4°C for a maximum of 7 days. Viral titre was determined by performing mock infections and quantifying fluorescent cells, via flow cytometry (MacsQUANT, Miltenyi Biotec Ltd.) 48 hr after infection. For both the genome-wide and focused-

**Table 2.** Examples of cancer-associated genetic vulnerabilities to G4 ligands.

| Gene category | Gene name | Function/pathway Summary | Cancer association summary | References |
|---|---|---|---|---|
| DNA damage repair | PALB2 | Homologous recombination; binds BRCA2 | Inactivating mutations predispose patients to myeloid leukaemia, Wilm's tumour and Fanconi anaemia. | *Harrigan et al., 2018*; *Nepomuceno et al., 2017* |
| | BAP1 | Homologous recombination; binds BRCA1, deubiquitinase for Histone 2A and tumour suppressor HCFC-1 | Inactivating mutations found in uveal melanoma and mesotheliomas. | *Harrigan et al., 2018*; *Carbone et al., 2013* |
| | USP1 | Fanconi anaemia and translesion synthesis DDR; deubiquitinase required for FANCD2, FANCI and PCNA localisation to sites of DNA damage | USP1 mRNA over-expressed in melanoma, gastric, cervical and NSCLC; under-expressed in leukaemia and lymphoma. | *Harrigan et al., 2018*; *Nijman et al., 2005*; *Huang et al., 2006*; *García-Santisteban et al., 2013* |
| | TOP1 | Relieves torsional stress during DNA replication; suppresses genomic instability at actively transcribed exogenous G4-forming sequences | Common cancer target, to induces DNA damage following pharmacological inhibition, lethal to cells. | *Yadav et al., 2014*; *Wang, 2002* |
| Helicase activity | RECQL4 RTEL1 | Previously identified G4-helicases | RECQL4 (Rothmund-Thomsun syndrome) and RTEL1 (Hoyeraal-Hreidarsson Syndrome), deficiencies impart increased risks of cancer cancer, autoimmunity and premature ageing. | *Brosh, 2013* |

*Table 2 continued on next page*

*Table 2 continued*

| Gene category | Gene name | Function/pathway Summary | Cancer association summary | References |
|---|---|---|---|---|
| Chromatin remodellers | ANKRD11 | Transcription factor; Recruits histone deacetylases | Tumour suppressor epigenetically silenced in breast cancers. | *Lim et al., 2012* *Neilsen et al., 2008* *Noll et al., 2012* |
| | MLL4 | H3K4 lysine methyl transferase | Frequently inactivated in several cancers. | *Froimchuk et al., 2017* *Kadoch et al., 2013;* *Rao and Dou, 2015* |
| | SMARCA4 SMARCB1 SMARCE1 | SWI/SNF ATP-dependent chromatin remodellers | Mutated in 20% of human cancers; doxorubicin resistant triple-negative breast cancer is associated with loss of SMARCB1, SMARCA4, or KEAP1 (a BRCA1 interactor). | *Kadoch et al., 2013;* *Shain and Pollack, 2013* |
| Ubiquitin | USP37 | Deubiquitinating enzyme which stabilises MYC | Upregulated in lung cancer. | *Pan et al., 2015* |
| | NEDD4L | E3 ubiquitin ligase | Expression correlates with poor patient outcome in hepatocellular and gastric carcinomas. | *Zhao et al., 2018;* *Gao et al., 2012* |
| | RNF20 | E3 ubiquitin ligase; chromatin remodelling and DDR | Tumour supressor down-regulated in several cancers. Deletion is main contributor to chromosomal instability in colorectal cancer. | *Moyal et al., 2011;* *Shema et al., 2008;* *Barber et al., 2008* |
| Splicing | FUS | Splicing component and known G4-interactor | Over-expressed in colon, breast and liposarcoma cancers, respectively. | *Crozat et al., 1993;* *Dvinge et al., 2016;* *Takahama et al., 2013* |

DOI: https://doi.org/10.7554/eLife.46793.025

screen, $3.6 \times 10^6$ target cells were infected with a viral volume predicted to cause 30% infection (MOI 0.3) to minimise multiple shRNA integrations per cell. This provides approximately $10 \times 10^6$ shRNA expressing cells (1000-fold shRNA representation). Virus was diluted in serum free media plus polybrene (8 µg/mL) with infections carried out in triplicate and treated as independent replicates hereafter. 48 hr after infection cells antibiotic selection was performed with 800 µg/mL (HT1080) and 1000 µg/mL (A375) geneticin for 7–9 days (antibiotic concentrations were determined from 7-day toxicity curves prior to transfection setup).

## Cell culture for pilot, genome-wide and focused shRNAs pools

Following complete antibiotic selection, a reference time point was harvested (t0) and cells were split into $3 \times 15$ cm plates per replica: PDS, PhenDC3 and DMSO vehicle control, each containing $8-10 \times 10^6$ cells to maintain 1000-fold hairpin representation. Every 72 hr, cells were trypsinised, counted via Muse Cell Analyzer to determine the number of population doublings, and $10 \times 10^6$ (A375 genome-wide and focused) or $8 \times 10^6$ (HT1080) cells per replica re-plated in fresh drug/DMSO and media (17 mL media per plate). At all times, sufficient cell numbers were used so that a

minimum of 1000 or 800 cells per shRNA was maintained (A375 and HT1080, respectively), to ensure maximal potential for uncovering phenotypic effects from each shRNA hairpin tested (*Knott et al., 2014*). The volume of DMSO used in the 'vehicle' condition is equal to the volume for 10 µM PhenDC3. The remaining drug treatments were supplemented with DMSO to match this volume to keep the same DMSO concentration between treatment cell lines and screens. For the pilot screen, two final timepoints were harvested after 7 and 15 population doublings (t7 and t15), pellets extracted and analysed as described below. Based on the pilot screen, discussed below, a final time point (tF) was harvested after 15 population doublings for subsequent genome-wide and focused screen. For each pool of the genome-wide screen, 12 samples were generated (t0, DMSO tF, PDS tF, PhenDC3 tF; three replicas each). Therefore, 144 samples of $10 \times 10^6$ cells were generated to cover the entire screen. For each cell line of the focused screen, 12 samples were generated (t0, DMSO tF, PDS tF, PhenDC3 tF, three replicas each).

## Pilot screen technique to determine genome-wide parameters

To determine the most appropriate tF for the genome-wide screen, cells (from shRNA pool 8) were harvested after 7 and 15 population doublings (t7 and t15 respectively) and the average $\log_2$FC (tF/t0) counts for each hairpin were determined as described below. For t7, 13 and 115 shRNAs were significantly altered following PDS and PhenDC3 treatment respectively (FDR $\leq$ 0.05). At t15, more hairpins were significantly depleted following PDS and PhenDC3 treatment (746 and 93 shRNAs respectively, excluding those significantly changed in DMSO).

## Barcode recovery, adapter ligation and sequencing

All PCR and sequencing oligonucleotides (Merck) are summarised in the table below. Cell pellets ($10 \times 10^6$ cells) were resuspended in PBS and genomic DNA extracted using QIAmp DNA Blood Maxi Kit (Qiagen, cat # 51194) according to the manufacturer's spin protocol, eluted in a final volume of 1200 µL and quantified by Qubit DNA HS Assay Kit (Thermo Fisher Scientific, cat # Q32851). The shRNA inserts were PCR-amplified from all DNA in each sample, in multiple 50 µL reactions each using 1.5 µg gDNA, with KOD Hot Start DNA Polymerase (Merck, cat # 710864) and the following reagents (included within the kit): 5 µL 10 x buffer, 5 µL 2 mM each dNTPs, 4 µL MgSO$_4$ (25 mM), 1.5 µL polymerase, 4 µL DMSO. Forward (Mir-F) and reverse (PGKpro-R) primers flanking the loop and antisense sequence of the hairpin region were used at a final concentration of 300 nM. PCR was performed under the following conditions: 98°C for 5 min, then for 25 cycles of 98°C for 35 s, 58°C 35 s and 72°C for 35 s, followed by a final extension at 72°C for 5 min. 1.2 mL of pooled PCR reaction were cleaned-up using QIAquick PCR purification kit (Qiagen, cat # 28104) according to manufacturer's protocol. 2 µg purified PCR product were PCR amplified in a second step, using forward (P5-Seq-P-Mir-Loop) and reverse (P7-Index-n-TruSeq-PGKpro-R) primers containing the P5 and P7 flow-cell adapters, respectively. PCR was performed in $8 \times 50$ µL reactions each with 500 ng template DNA. The reverse primer contains TruSeq adapter small RNA Indexes for multiplexing and a 6-nucleotide barcode, denoted 'nnnnnn' below. PCR reagents were as for the first PCR, with the exception of the primers, which were used at a final concentration 1.5 µM. The second PCR was performed under the following conditions: 98°C for 5 min, then for 25 cycles of 94°C for 35 s, 52°C 35 s and 72°C for 35 s, followed by a final extension at 72°C for 5 min. 200 µL of pooled secondary PCR product was cleaned up as previously and the desired product (~340 bp) was extracted using BluePippin (Sage Science) 2% Internal Standard Marker Kit (DF marker 100–600 bp; Sage Science, BDF2010), according to manufacturer's protocol using a broad range elution (300–400 bp). Individual samples were quantified with a KAPA library quantification kit (KAPA Biosystems, cat # 0796-6014-0001) using a BioRad CFX96 Real Time PCR instrument with no Rox according to manufacturer's protocols. Libraries were diluted to 4 nM in RNAse free water. For the genome-wide screen samples, 24 libraries (12 pools) and for the focused screen samples, 24 libraries (both cell lines) were combined to create a pooled 4 nM stock, with each sample having a unique TruSeq adapter. The genome-wide screen samples were sequenced in six batches; all focused screen samples were sequenced simultaneously. DNA-Seq libraries were prepared from these samples using the NextSeq Illumina Platform v2 High Output Kit 75 cycles, followed by 36 base pair single-read sequencing performed on an Illumina NextSeq instrument, using a custom sequencing primer.

| Oligo name | Description | Sequence 5'−3' |
|---|---|---|
| Mir5-F | Primary PCR Forward Primer | 5'-CAGAATCGTTGCCTGCACATCTTGGAAAC- 3' |
| PGKpro-R | Primary PCR Reverse Primer | 5' -CTGCTAAA GCGCATGCTCCAGACTGC- 3' |
| P5-Seq-P-Mir-Loop | Secondary PCR forward Primer | 5'-AATGATACGGCGACCACCGAGATCTACACT AGCCTGCGCACGTAGTGAAGCCACAGATGTA-3' |
| P7-Index-n-Truseq-PGKpro-R | Secondary PCR barcoded reverse primer | 5'-CAAGCAGAAGACGGCATACGAGAT nnnnnnGTGACTGGAGTTCAGACGTGTGCTCTT CCGATCTCTGCTAAAGCGCATGCTCCAGACTGC – 3' |
| SeqPrimer MirLoop | Custom sequencing primer | 5'-TAGCCTGCGCACGTAGTGAAGCCACAGATGTA-3 |

## Sequencing, read processing, alignment and counting of shRNAs

Sequencing data have been deposited in ArrayExpress (https://www.ebi.ac.uk/arrayexpress/) under accession number *E-MTAB-6367*. Reads were trimmed to 22 nucleotides, base qualities were evaluated with FastQC v0.11.3 (*Andrews, 2010*) and bases were filtered from the 3' end with a Phred quality threshold of 33 using the FASTX-Toolkit v0.0.14 (*Gordon and Hannon, 2010*). Trimmed reads were aligned to the 113,002 reference shRNA sequences provided by transOMIC technologies (*Knott et al., 2014*) using Bowtie 2 v2.2.6 with default parameters (*Langmead and Salzberg, 2012*), which resulted in overall alignment rates of 90–95% with an average of 98% of reference sequences detected. The generated SAM files were processed to obtain shRNA counts using Unix tools (https://opengroup.org/unix) and Python scripts (v2.7.10, https://www.python.org), and library purity and potential contaminations were investigated with stacked bar plots and multidimensional scaling (MDS) using the R programming language v3.2.1 (https://cran.r-project.org). The code and scripts developed during the development of the project are available in our group's GitHub website (*Martínez Cuesta, 2019*; copy archived at https://github.com/elifesciences-publications/GWscreen_G4sensitivity).

## Filtering, normalisation, differential representation analysis and defining sensitisation

To discard shRNAs bearing low counts, each library was filtered based on a counts-per-million threshold of 0.5 for all initial time points (t0), for example in a library of 10M reads, shRNAs with at least five counts for all initial time points will pass this filter. Normalisation factors were calculated to scale the raw library sizes using the weighted trimmed mean of M-values (TMM) approach (*Robinson et al., 2010*). To compare groups of replicates (time points and chemical treatments) for each pool, differential representation analysis of shRNA counts was performed using edgeR (*Robinson et al., 2010*). Common and shRNA-specific dispersions were estimated to allow the fitting of a negative binomial generalised linear model to the treatment counts. Contrasts between the initial time point and the treatments were defined (PDS-t0, PhenDC3-t0, and DMSO-t0) and likelihood ratio tests were carried out accordingly (*Dai et al., 2014*). Fold changes (FC) were then computed for every shRNA, and false discovery rates (FDR) were estimated using the Benjamini-Hochberg method. A gene was defined as significantly differentially represented for a given treatment if at least 50% or a minimum of 3 shRNAs were significant (FDR $\leq$ 0.05); sensitisation was additionally determined by applying a $\log_2 FC \leq -1$.

## Exploring genes associated to G4s in databases and biomedical literature

Three different approaches were developed to uncover genes linked to G4s in the literature and molecular biology databases. 18 high confidence G4-related genes were obtained by scanning for genes in which the corresponding UniprotKB (*The UniProt Consortium, 2017*) entry is annotated with the term 'quadruplex' or genes annotated with at least one of the following 11 GO terms with any evidence assertion method (*Ashburner et al., 2000*):

| GO id | Name | Type | Link |
|-------|------|------|------|
| GO:0051880 | G-quadruplex DNA binding | Molecular function | https://www.ebi.ac.uk/QuickGO/term/GO:0051880 |
| GO:0002151 | G-quadruplex RNA binding | Molecular function | https://www.ebi.ac.uk/QuickGO/term/GO:0002151 |
| GO:0061849 | telomeric G-quadruplex DNA binding | Molecular function | https://www.ebi.ac.uk/QuickGO/term/GO:0061849 |
| GO:0071919 | G-quadruplex DNA formation | Biological process | https://www.ebi.ac.uk/QuickGO/term/GO:0071919 |
| GO:0044806 | G-quadruplex DNA unwinding | Biological process | https://www.ebi.ac.uk/QuickGO/term/GO:0044806 |
| GO:1905493 | regulation of G-quadruplex DNA binding | Biological process | https://www.ebi.ac.uk/QuickGO/term/GO:1905493 |
| GO:1905494 | negative regulation of G-quadruplex DNA binding | Biological process | https://www.ebi.ac.uk/QuickGO/term/GO:1905494 |
| GO:1905495 | positive regulation of G-quadruplex DNA binding | Biological process | https://www.ebi.ac.uk/QuickGO/term/GO:1905495 |
| GO:1905465 | regulation of G-quadruplex DNA unwinding | Biological process | https://www.ebi.ac.uk/QuickGO/term/GO:1905465 |
| GO:1905466 | negative regulation of G-quadruplex DNA unwinding | Biological process | https://www.ebi.ac.uk/QuickGO/term/GO:1905466 |
| GO:1905467 | positive regulation of G-quadruplex DNA unwinding | Biological process | https://www.ebi.ac.uk/QuickGO/term/GO:1905467 |

Furthermore, 55 confirmed human G4-interacting proteins as defined by the G4IPB database (*Mishra et al., 2016*) (http://bsbe.iiti.ac.in/bsbe/ipdb/index.php) were also used to determine predefined G4-interacting proteins from the genome-wide shRNA screen. For this, gene entries were removed where the only G4-relationship was a predicted G4-forming sequence in the mRNA or DNA (i.e. not a direct protein interaction) or where the protein was not human. To extend the list of G4-interacting proteins, text-mining using PolySearch2 (*Liu et al., 2015b*) was used. Human protein-coding gene names and G4-terms and synonyms are defined using the corresponding MeSH term id D054856 (https://www.ncbi.nlm.nih.gov/mesh/68054856) and the thesaurus of gene names obtained from the PolySearch2 website (http://polysearch.cs.ualberta.ca/). A relevancy score measures the strength of association between two text groups, and higher the score, the more likely terms from the two groups co-occur within the same abstract; the score also accounts for the distance between terms from the two groups. A total of 5477 pieces of text were identified in PubMed and PubMed Central where any of the G4 terms co-occur with more than 500 human gene names. Overall, this generated 526 G4-associated genes, with 54 (10%) uncovered as G4-sensitisers (https://github.com/sblab-bioinformatics/GWscreen_G4sensitivity), which were manually edited to 16 genes as discussed in the main text and figures.

### KEGG pathway, gene ontology and protein domain enrichment analysis

The ClueGO v2.3.3 (*Bindea et al., 2009*; *Bindea et al., 2013*) plugin for Cytoscape (*Shannon et al., 2003*) (v3.5.1) was used to determine networks of enriched KEGG pathways and Gene Ontology terms (Biological Process and Molecular Function) for significantly depleted genes upon G4 ligand

treatment. Specifically, a right-sided (Enrichment) test based on the hyper-geometric distribution was performed on the corresponding Entrez gene IDs for each gene list and the Bonferroni adjustment (p<0.05) was applied to correct for multiple hypothesis testing. Only experimental evidence codes (EXP, IDA, IPI, IMP, IGI, IEP) were used. The Kappa-statistics score threshold was set to 0.4 and GO term fusion was used to diminish redundancy of terms shared by similar proteins. Other parameters include: GO level intervals (3–8 genes) and Group Merge (50%). Protein domains were investigated using DAVID (v6.8) to integrate GENE3D crystallographic data and PFAM sequence information and enrichment was considered significant if the EASE score p<0.05 (*Finn et al., 2016*; *Yeats et al., 2006*).

## COSMIC analysis

Cancer mutation data (CosmicMutantExport.tsv) from the COSMIC database v82 (*Forbes et al., 2015*) was used to investigate the association between G4 sensitisers and cancer genes. ~150,000 were mutations available in COSMIC for 702 (93%) sensitiser genes, with some predicted to be pathogenic by the FATHMM algorithm embedded within the COSMIC database. The Cancer Gene Census (http://cancer.sanger.ac.uk/census) was used to investigate whether G4 sensitisers are enriched in genes containing mutations causally implicated in cancer. Fisher's exact tests as implemented in R were used to calculate fold enrichment significance of sensitisers that are cancer genes in COSMIC (compared to the percentage of protein-coding genes in COSMIC – 3.3%).

## siRNA validation experiments – transfection, experimental outline, immunoblotting

ON-TARGETplus siRNAs (Dharmacon/GE healthcare) were used as summarised in the table below. Cells were transfected with either targeting or non-targeting control siRNAs using lipofectamine RNAiMAX (Thermo Fisher Scientific, cat # 13778150) and OptiMEM reduced serum medium (Thermo Fisher Scientific) according to the manufacturer's protocol (reagent protocol 2013) alongside a non-transfected control. 24 hr after transfection, cells were trypsinised, counted and re-plated in media supplemented with PDS, PhenDC3 or DMSO vehicle control (minimum two biological replicates per condition) in a 48-well plate format (seeding density - 8,000 cells per well A375; 4,000 cells per well HT1080). Cell growth was monitored for 144 hr using IncuCyteZOOM live cell analysis (Sartorius) and cell confluency calculated as a percentage of the well area covered. Scans were performed every 3 hr; nine scans per well. To monitor protein levels, cells transfected simultaneously with the same siRNA-reagent mixture were harvested 48 hr and 144 hr after transfection, by cell scraping and lysed on ice (30 min) with RIPA lysis buffer with protease inhibitor +EDTA (Thermo Fisher Scientific, cat # 8990). Lysates were analysed by capillary electrophoresis via the Protein Simple Wes platform according to manufacturer's protocol with antibodies summarised in the key resource table above. Lysates from non-transfected and siRNA-treated (targeting and non-targeting) samples were probed with antibodies against BRCA1 (Cell Signalling Technology, cat # 4970-CST), TOP1 (Abcam, cat # AB109374), GAR1 (NovusBio cat #NBP2-31742) or DDX42 (Abcam cat #AB80975), plus anti-beta actin antibody (mouse Merck cat # A5441; rabbit cat # 4970-CST) by multiplexing. For non-targeting and targeting lysates, the area of the desired band was normalised to beta-actin and then normalised to the protein level in the non-targeting sample, for three (48 hr after transfection lysates) or two independent Wes runs (144 hr after transfection). Protein depletion is expressed as an average of these normalised values. All lysates were used at a concentration of 0.8 mg/mL and antibody dilutions as follows: BRCA1 1:50; TOP1 1:250; GAR1 1:100; DDX42 1:250; rabbit-actin 1:500; mouse-actin 1:250.

| siRNA | Catalogue number | Sequence 5'—3' |
|---|---|---|
| Non-targeting 2 | D-001810-02-05 | UGGUUUACAUGUUGUGUGA |
| BRCA1 (A375) | J-003461–09 | CAACAUGCCCACAGAUCAA |
| BRCA1 (HT1080) | J-003461–12 | GAAGGAGCUUUCAUCAUUUC |
| TOP1 (both cell lines) | J-005278–08 | CGAAGAAGGUAGUAGAGUC |
| DDX42 (both cell lines) | J-012393–11 | GGAGAUCGACUAACGGCAA |
| GAR1 (both cell lines) | J-013386–06 | UCCAGAACGUGUAGUCUUA |

## G4 ligand and drug treatments

10 mM stocks in DMSO of PDS (in house synthesis), MK1775 (Cambridge Bioscience, cat# CAY21266) and pimozide (Merck, cat# P1793-500MG) were used as for synergy experiment. Cells were seeded in Corning, Tissue Culture-treated 96-well clear bottom plates (Thermofisher, cat#07-200-587) for HT1080 (1000 cells per well) and A375 (1500 cells per well) cell lines. 24 hr after plating, media was removed and cells were treated with different concentrations of PDS and MK1775 or pimozide in media in a final volume of 150 µL, alongside non-treated and solvent-treated controls. After 96 hr, cell death was determined via a CellTitre-Glo One Solution assay (Promega, cat # G8461) according to manufacturer's protocol using the PHERAstar FS (BMG labtech) to detect luminescence, using the recommended settings. Values were normalised to and expressed as a percentage of the untreated controls. This was performed for three biological replicas. Data were analysed via Combenefit software using the BLISS independence model since the molecule have independent targets (*Di Veroli et al., 2016*) to determine synergy. For the clonogenic cell survival assay, A375 (300 cells per well) and HT1080 (400 cells per well) were plated as single cells in 12-well plates. The next day, cells were treated with DMSO or the indicated doses of PDS, pimozide and/or MK1775 in media. After 8 days, colonies were fixed with 3% trichloroacetic acid (TCA) for 90 min at 4°C, washed with MiliQ, air dried and then stained with 0.057% (v/v) Sulforhodamine B solution (Merck, cat # 230162–5G) for 30 min at room temperature. Plates were then washed four times with 1% acetic acid, air dried and colonies visualised using GelCount (Oxford Optronix). Colony growth was determined using the 'colony intensity percentage' parameter in the ColonyArea Image J plugin (*Guzmán et al., 2014*), which considers both the intensity and percentage of area covered by the colonies. Values were normalised to and expressed as a percentage of the untreated controls and then further processed by Combenefit software, as described above, to determine synergy. A total of three independent biological replicates were performed.

## Sub-cellular localisation of DDX42

HT1080, HEK and HeLa cells were harvested from a 70% confluent 15 cm plate by cell scraping in PBS on ice and pelleted by centrifugation (500 g, 5 min, 4°C). Pellets were resuspended in three volumes of low-salt buffer (20 mM HEPES pH7.4, 10 mM NaCl, 3 mM $MgCl_2$, 0.2 mM EDTA, 1 mM DTT) plus protease inhibitor (cOmplete mini, Roche cat#11836153001), lysed on ice (15 min) and 0.5% Igepal added. Samples were vortexed (1 min) and centrifuged (900 g, 15 min, 4°C) and the supernatant collected for cytoplasmic extracts. Nuclear pellets were washed in low-salt buffer, supernatant discarded and then resuspended in high-salt buffer (20 mM HEPES pH7.4, 500 mM NaCl, 3 mM $MgCl_2$, 0.5% Igepal, 0.2 mM EDTA, 1 mM DTT) plus protease inhibitors, followed by lysis on ice with intermittent vortexing (30 min). Lysates were passed through a syringe needle to promote lysis and shear genomic DNA and followed by centrifugation (13,000 g, 10 min, 4°C). Lysis confirmed by trypan blue staining according to manufacturer's protocols (Thermofisher Scientific cat#15250061). The supernatant was then collected as nuclear extract. Cytoplasmic and nuclear lysates were quantified on a Direct Detect platform (Merck) and DDX42 expression analysed by immunoblotting using the Protein Simple Wes instrument as described above with a lysate concentration of 0.5 mg/mL. Samples were also immunoblotted with antibodies against nuclear laminB1 (CST 12586; 1:250) and cytoplasmic GAPDH to confirm subcellular fractionation efficiency (CST 5174, 1:50).

## Oligonucleotide annealing

Biotinylated oligonucleotides for G4 and non-G4 forming sequences (IDT technologies; see Table below) were annealed in 10 mM TrisHCl pH 7.4, 100 mM KCl by heating at 95°C, 10 min followed by slow cooling to room temperature overnight at a controlled rate of 0.2°C/min. Annealed oligonucleotides were stored at 4°C for maximum 1 month.

| Oligo | Rna/DNA | Sequence |
|---|---|---|
| NRAS G4 | RNA | 5' [Btn] UGU GGG AGG GGC GGG UCU GGG UGC 3' |

*Continued on next page*

*Continued*

| Oligo | Rna/DNA | Sequence |
| --- | --- | --- |
| NRAS mut | RNA | 5' [Btn] UGU AGA AAG AGC AGA UCU AGA UG 3' |
| Stem loop | RNA | 5' [Btn] ACA GGG CUC CGC GAU GGC GGA GCC CAA 3' |
| Myc G4 | DNA | 5' [Btn] TGA GGG T GGG TA GGG T GGG TAA 3' |
| Myc mut | DNA | 5' [Btn] TGA GAG T GAG TA GAG T GAG TAA 3' |

## Enzyme-Linked immunosorbent assay

Recombinant human DDX42 with an N-terminal GST tag was purchased from NovusBio (cat# H0001325-P01). Streptavidin-Coated High-Binding Capacity 96-well plates (ThermoScientific prod #15501) were hydrated with PBS (30 min) and coated with 50 nM biotinylated oligonucleotides (1 hr, shaking 450 rpm). Wells were washed three times with ELISA buffer (50 mM $K_2HPO_4$ pH 7.4 and 100 mM KCl/100 mM LiCl); 1 min shaking, 450 rpm. Wells were then blocked with 3% (w/v) BSA (Merck, cat# A7030) in ELISA buffer for 1 hr, at room temperature and then incubated with serial dilutions of DDX42 up to 200 nM for 1 hr. Wells were washed three times with 0.1% TWEEN-20 in ELISA buffer and then incubated for 1 hr with anti-GST HRP-conjugated antibody (Abcam AB3416) diluted 1:10,000 in blocking buffer. Wells were again washed three times with ELISA-Tween, and the bound anti-GST HRP detected with TMB substrate (Merck,cat#T4444) for 2 min. Reactions were stopped with 2 M HCl. Absorbance at 450 nm was measured with PheraSTAR plate reader (BMG labtech). Binding curves with standard error of the mean (SEM) were fitted using GraphPad Prism software, using a non-linear regression fit, one site, specific binding model with saturation kinetics. The following equation was used: $y=(Bmax*x)/(K_d +x)$, where x = concentration of DDX42 (nM) and Bmax is the maximum specific binding (i.e. saturation).

## Circular dichroism spectroscopy

200 µL of 10 µM oligonucleotide were prepared in assay buffer and annealed as described above. CD spectra were recorded on an Applied Photo-physics Chirascan CD spectropolarimeter using a 1 mm path length quartz cuvette. CD measurements were performed at 298 K over a range of 200–320 nm using a response time of 0.5 s, 1 nm pitch and 0.5 nm bandwidth. The recorded spectra represent a smoothed average of three scans, zero-corrected at 320 nm (Molar ellipticity $\theta$ is quoted in $10^5$ deg $cm^2$ $dmol^{-1}$). The absorbance of the buffer was subtracted from the recorded spectra.

## Acknowledgements

The Balasubramanian laboratory is supported by core funding (C14303/A17197) and programme grant funding (C9681/A18618) from Cancer Research UK. SB is a Senior Investigator of the Wellcome Trust (grant no. 209441/z/17/z). GJH is supported by core funding from Cancer Research UK (A21143) and a Royal Society Research Professorship (RP130039). NE was supported by a grant from the Pew Charitable Trusts (00028354). Authors would like to thank Dr Isaac Johnson for help regarding the clonogenic survival assays. Authors would also like to thank the staff of the Genomic, Flow Cytometry, Research Instrumentation and Biorepository core facilities at CRUK Cambridge Institute and members of the Balasubramanian and Hannon Lab for helpful discussions.

## Additional information

### Competing interests

Gregory J Hannon: Is associated with transOMIC Technologies, who have commercialised libraries constructed using the shERWOOD and ultramiR design strategies. Shankar Balasubramanian: Is founder, adviser and shareholder of Cambridge Epigenetix Ltd. The other authors declare that no competing interests exist.

## Funding

| Funder | Grant reference number | Author |
|---|---|---|
| Cancer Research UK | C14303/A17197 | Katherine G Zyner<br>Darcie S Mulhearn<br>Santosh Adhikari<br>Marco Di Antonio<br>David Tannahill<br>Shankar Balasubramanian |
| Cancer Research UK | C9681/A18618 | Katherine G Zyner<br>Darcie S Mulhearn<br>Santosh Adhikari<br>Marco Di Antonio<br>David Tannahill<br>Shankar Balasubramanian |
| Cancer Research UK | A21143 | Gregory J Hannon |
| Royal Society | RP130039 | Gregory J Hannon |
| Pew Charitable Trusts | 00028354 | Nicolas Erard |
| Wellcome | 209441/z/17/z | Sergio Martínez Cuesta<br>Shankar Balasubramanian |

The funders had no role in study design, data collection and interpretation, or the decision to submit the work for publication.

## Author contributions

Katherine G Zyner, Conceptualization, Data curation, Formal analysis, Supervision, Validation, Investigation, Visualization, Methodology, Writing—original draft, Writing—review and editing, Designed and performed the genome-wide shRNA screen, library preparation and sequencing, Designed the focused RNAi screens, Performed the GO and Cytoscape analyses, Designed and performed the clonogenic assays and DDX42 binding experiments; Darcie S Mulhearn, Conceptualization, Data curation, Formal analysis, Validation, Investigation, Visualization, Methodology, Writing—original draft, Writing—review and editing, Designed and performed the genome-wide shRNA screen, library preparation and sequencing, Designed and performed the focused RNAi screens, Performed the GO and Cytoscape analyses, Designed and performed the drug synergy and DDX42 binding experiments; Santosh Adhikari, Conceptualization, Data curation, Formal analysis, Validation, Investigation, Visualization, Methodology, Writing—original draft, Writing—review and editing, Designed and performed the drug synergy experiments; Sergio Martínez Cuesta, Conceptualization, Data curation, Software, Formal analysis, Validation, Methodology, Writing—original draft, Writing—review and editing, Performed the computational analyses on all RNAi screens; Marco Di Antonio, Conceptualization, Resources, Data curation, Formal analysis, Methodology, Writing—original draft, Writing—review and editing, Synthesised all the G4 ligands; Nicolas Erard, Resources, Software, Formal analysis, Methodology, Writing—review and editing; Gregory J Hannon, Conceptualization, Resources, Supervision, Methodology, Writing—original draft, Writing—review and editing; David Tannahill, Conceptualization, Resources, Software, Formal analysis, Supervision, Methodology, Writing—original draft, Project administration, Writing—review and editing; Shankar Balasubramanian, Conceptualization, Formal analysis, Supervision, Funding acquisition, Methodology, Writing—original draft, Project administration, Writing—review and editing

## Author ORCIDs

Katherine G Zyner  https://orcid.org/0000-0002-4997-0150
Santosh Adhikari  https://orcid.org/0000-0002-1501-2106
Sergio Martínez Cuesta  https://orcid.org/0000-0001-9806-2805
Marco Di Antonio  https://orcid.org/0000-0002-7321-1867
Gregory J Hannon  https://orcid.org/0000-0003-4021-3898
David Tannahill  https://orcid.org/0000-0002-3811-6864
Shankar Balasubramanian  https://orcid.org/0000-0002-0281-5815

Decision letter and Author response
Decision letter https://doi.org/10.7554/eLife.46793.032
Author response https://doi.org/10.7554/eLife.46793.033

## Additional files

### Supplementary files

• Supplementary file 1. Supporting data for *Figure 3–7*. List of shRNAs/genes from venn diagrams and table statistics for KEGG, GO, DGIDb and Protein Domains analyses from *Figures 3–7*. Each data tab is labelled with its corresponding originating figure.
DOI: https://doi.org/10.7554/eLife.46793.026

• Supplementary file 2. Supporting data for *Figure 6—figure supplement 1* and *Figure 7—figure supplement 1*. List of shRNAs/genes from venn diagrams and table statistics for GO analyses from *Figure 6—figure supplement 1* and *Figure 7—figure supplement 1*. Each data tab is labelled with its corresponding originating figure.
DOI: https://doi.org/10.7554/eLife.46793.027

• Transparent reporting form
DOI: https://doi.org/10.7554/eLife.46793.028

### Data availability

Sequencing data have been deposited in ArrayExpress under the accession number E-MTAB-6367.

The following dataset was generated:

| Author(s) | Year | Dataset title | Dataset URL | Database and Identifier |
|---|---|---|---|---|
| Mulhearn DS, Zyner KG, Martinez Cuesta S, Balasubramanian S | 2019 | Systematic identification of G-quadruplex sensitive lethality by genome-wide genetic screening | https://www.ebi.ac.uk/arrayexpress/experiments/E-MTAB-6367/ | ArrayExpress, E-MTAB-6367 |

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
