## [Decision Letter]

Thank you for submitting your article "Genetic Interactions of G-quadruplexes" for consideration by *eLife*. Your article has been reviewed by three peer reviewers, including Wolf-Dietrich Heyer as the Reviewing Editor and Reviewer #1, and the evaluation has been overseen by Kevin Struhl as the Senior Editor. The following individual involved in review of your submission has agreed to reveal their identity: Peter M Lansdorp (Reviewer #3).

The reviewers have discussed the reviews with one another and the Reviewing Editor has drafted this decision to help you prepare a revised submission.

Summary:

The authors report a genome-wide shRNA screen for gene knockdowns that sensitize cells to two G4 stabilizing compounds, Phen DC3 and pyridostatin (PDS), in the human melanoma cell line A375. They design a custom library of positive hits and shRNAs against additional G4-related genes and conduct two additional focused screens in A375 and in HT1080 fibrosarcoma cells. The authors identify 36 genes that are consistently positive in all screens for both G4 stabilizing compounds. Four genes, BRCA1, TOP1, DDX42, and GAR1, are independently validated by siRNAs in both cell lines. Further experiments focus on two druggable targets, WEE1 and USP1, and targeted inhibitors (WEE1: MK1775; USP1: pimozide) are found to synergize with PDS in both cell lines to various degrees. Moreover, they identified a novel gene, DXX42, and determine this RNA helicase to be a G4 RNA binding protein using recombinant protein.

This is a solid study linking many previously unknown genes to G4 biology as studied using G4 stabilizers. The fact that many genes in different cell types were identified with two chemically distinct G4 stabilizers and that, based on this, pharmacological inhibition of selected pathways was shown to be effective is a wonderful illustration of insightful biological experimentation with potential implications for novel cancer therapies.

After addition of some needed experimentation and expanded discussion of results, as outlined under essential revisions, this manuscript should make a valuable contribution and a 'Tools and Resources' article.

Essential revisions:

Points 1 and 2 require new experimentation.

1) Survival data should also include a clonogenic survival assay at least in selected examples such as Figure 9.

2) The experiments reported in Figure 10F and G are only with n=2, which is insufficient, and an 'n' of 3 or higher is required to derive such quantitative data.

Points 3-5 do not require additional experimentation but expanded discussion.

3) Do the genes that are differentially affected by the different G4 stabilizers (PDS and PhenDC3) show any similarity in G4 motifs in the gene body, promotor or nearby enhancers? Can anything be learned about G4 biology by comparing G4 motifs in the vicinity of genes that are 1) picked up by both G4 stabilizers or those identified by individual G4 stabilizers?

4) Is the growth inhibition observed with specific combinations of G4 stabilizers and shRNAs biologically relevant? If G4 formation and stabilization is sporadic, many cells could potentially escape the "synthetic lethal" interactions. Perhaps "significant G4-ligand mediated changes" is a more appropriate description of effects on cell growth than "synthetic lethality". For interpretation of potential clinical relevance, it is important that reader can distinguish minor growth inhibition (presumably with little clinical relevance) to robust killing of cells. This needs to be flashed out more e.g. by comparisons with previously described synthetic lethal combinations such as e.g. PARP inhibition in BRCA1 or 2 deficient cells.

5) Genetic abnormalities in the target cells for the initial shRNA screen using A375 melanoma cells need to be spelled out. A similar screen using HT1080 cells identified many G4 ligand sensitizers (42% of genes) that were not shared with those seen the A375 genome-wide screen. The authors should expand their discussion of possible explanations for these observations.

6) Figure 8C-F shows growth responses to two doses of each drug in the validation experiments with siRNA for BRCA1, TOP1, DDX42, and GAR1 in HT1080 cells. Why is there no dose response? Is there any explanation for the differences between PDS and PhenDC3? How can the differences between HT1080 and A375 cells (Figure 8—figure supplement 3) be explained.

7) The observation that out of 9509 G4-ligand specific shRNAs drop-outs in the screen only 843 or 758 identify the same gene suggest that the screen picks up many non-specific "synthetic lethal" hits. This deserves further explanation by the authors. The list of these genes that are supported by multiple shRNAs is nevertheless of interest and it is reassuring that several of the genes previously implicated in G4 biology and HR are detected.

---

## [Author Response]

Essential revisions:Points 1 and 2 require new experimentation.1) Survival data should also include a clonogenic survival assay at least in selected examples such as Figure 9.

We have performed the requested clonogenic survival assay for the examples given in Figure 9 and the results confirm our previous observations. We note that the clonogenic assay is more sensitive than the shorter-term cell killing assay necessitating lower compound concentrations. We have added a comment to the subsection “G4-targeting ligands plus pharmacological inhibitors of G4 sensitiser genes demonstrate synergistic cell killing”, and these data in Figure 9—figure supplement 2.

2) The experiments reported in Figure 10F and G are only with n=2, which is insufficient, and an 'n' of 3 or higher is required to derive such quantitative data.

We obtained a new batch of recombinant protein and performed repeat experiments for a total of five replicates which gives similar results. The updated data are presented in Figure 10.

Points 3-5 do not require additional experimentation but expanded discussion.3) Do the genes that are differentially affected by the different G4 stabilizers (PDS and PhenDC3) show any similarity in G4 motifs in the gene body, promotor or nearby enhancers? Can anything be learned about G4 biology by comparing G4 motifs in the vicinity of genes that are 1) picked up by both G4 stabilizers or those identified by individual G4 stabilizers?

These are interesting questions, but they cannot be addressed owing to the very nature our screen, which actually determines cell survival in the presence of a G4 ligand in the event where expression of a given gene is independently knocked-down by shRNA. The outcome does not depend on whether or not the G4 ligand interacts directly with the knocked-down gene.

4) Is the growth inhibition observed with specific combinations of G4 stabilizers and shRNAs biologically relevant?

Yes, we believe that many will be biologically relevant. The biology of G4s is an emerging field and their detailed functions are largely unknown, but we and others have shown that loss of specific genes, several of which were identified in our study, in the presence of G4 ligands leads to demonstrable biological effects (subsection “Identification of genetic vulnerabilities to G4-ligands via genome-wide screening”, third paragraph). Furthermore, as referenced in the Introduction (fourth paragraph), we and others have shown that cells deficient in homologous recombination are more susceptible to G4 ligands. Importantly, the latter findings were used as the basis for development of a G4 ligand, that we have helped characterise (Xu et al., 2017), that has entered clinical trials for BRCA-deficient breast cancer patients (NCT02719977 ClinicalTrials.gov).

If G4 formation and stabilization is sporadic, many cells could potentially escape the "synthetic lethal" interactions.

It is commonly thought that G4s are persistent structures. Supporting this, we and others using antibody tools have demonstrated that G4s form consistently (see Introduction, second paragraph). Even if a proportion of G4s formed sporadically, they would have to be processed by the cell during transcription and replication, so defects in G4 unwinding enzymes would still be apparent and our screen did indeed uncover many such proteins.

Perhaps "significant G4-ligand mediated changes" is a more appropriate description of effects on cell growth than "synthetic lethality".

We have accepted that our use of the terminology “synthetic lethality” may have been confusing and have clarified this throughout the manuscript.

For interpretation of potential clinical relevance, it is important that reader can distinguish minor growth inhibition (presumably with little clinical relevance) to robust killing of cells. This needs to be flashed out more e.g. by comparisons with previously described synthetic lethal combinations such as e.g. PARP inhibition in BRCA1 or 2 deficient cells.

We believe that our observations can be the starting point for future clinical work (see our first response to point 4). PDS and PhenDC3 are tool molecules and have not undergone extensive medicinal chemistry optimisation. When one compares PDS to the clinically developed CX5461 G4 ligand, both show robust cellular effects though CX5461 has greater efficacy (Xu et al., 2017). Furthermore, Zimmer et al. directly compared the efficacy of PDS and olaparib in BRCA-deficient cells and found that both molecules had comparable growth inhibitory properties (Zimmer et al., 2016). We have added further discussion on the comparisons on PDS and olaparib, and also robust vs. minor growth effects in the second paragraph of the Discussion.

5) Genetic abnormalities in the target cells for the initial shRNA screen using A375 melanoma cells need to be spelled out.

We have added comments on key genetic abnormalities for these cell lines. See subsection “Identification of genetic vulnerabilities to G4-ligands via genome-wide screening”, first paragraph and subsection “BRCA1, TOP1, DDX42 and GAR1 are key G4 ligand sensitiser genes”.

A similar screen using HT1080 cells identified many G4 ligand sensitizers (42% of genes) that were not shared with those seen the A375 genome-wide screen. The authors should expand their discussion of possible explanations for these observations.

We have added relevant discussion of these points in the fifth paragraph of the Discussion.

6) Figure 8C-F shows growth responses to two doses of each drug in the validation experiments with siRNA for BRCA1, TOP1, DDX42, and GAR1 in HT1080 cells. Why is there no dose response?

We have added text for this in the fifth paragraph of the Discussion.

Is there any explanation for the differences between PDS and PhenDC3?

See our second response to point 5. Also, PhenDC3 and PDS are structurally unrelated compounds, thus they have different physico-chemical properties and different binding properties towards G4s, which themselves represent heterogeneous targets. For example, our own work has shown that in addition to having different binding affinities, PhenDC3, unlike PDS, has a strong preference for 5’-G-tetrad binding (Le et al., 2015). We have added a discussion on this to the fifth paragraph of the Discussion.

How can the differences between HT1080 and A375 cells (Figure 8—figure supplement 3) be explained.

We suspect that the differences between the cell lines are in part due to the greater knockdown efficiency seen with HT1080 cells compared to A375 cells. We have added a comment for this to the fifth paragraph of the Discussion.

7) The observation that out of 9509 G4-ligand specific shRNAs drop-outs in the screen only 843 or 758 identify the same gene suggest that the screen picks up many non-specific "synthetic lethal" hits. This deserves further explanation by the authors. The list of these genes that are supported by multiple shRNAs is nevertheless of interest and it is reassuring that several of the genes previously implicated in G4 biology and HR are detected.

Thank you for pointing this out. Although the efficacy of genome-wide screening libraries has improved considerably with the development of new algorithms for library design (reviewed in 10.1038/nrg.2017.47), the lack of complete gene specificity of shRNA hairpins (and CRISPR sgRNAs) still hinders the outcome of genome-wide studies. It is standard practice in shRNA screens that the desired phenotype is captured by multiple independent shRNA hairpins to mitigate any off-target effects and variability in knockdown efficiencies (reviewed in 10.1038/nrg2364, 10.1038/nmeth.1351 and 10.1038/nmeth1006-777). To clarify further, the 9,509 G4 ligand specific shRNA hairpins arise from the shRNA cut-off threshold (FDR <0.05) alone whereas the 758 G4 synthetic lethal genes (representing 2,025 unique shRNA hairpins) come from additionally considering a gene threshold cut-off (3 or 50% hairpins and median Log_2_FC ≤ -1). 5411 genes are targeted by a single hairpin whereas 2329 genes are targeted by two or more hairpins. We have added an extra table (Figure 3 shRNA_summary) into Supplementary file 1 to provide the information.